# CED-5/CED-12 (DOCK/ELMO) can promote and inhibit F-actin formation via distinct motifs that may target different GTPases

**Thejasvi Venkatachalam, Sushma Mannimala, Yeshaswi Pulijala®, Martha C. Soto®***

Department of Pathology and Laboratory Medicine, Rutgers–Robert Wood Johnson Medical School, Piscataway, New Jersey, United States of America

* sotomc@rutgers.edu

**Data Availability Statement:** All relevant data are within the manuscript and its Supporting Information files.

## Abstract

Coordinated activation and inhibition of F-actin supports the movements of morphogenesis. Understanding the proteins that regulate F-actin is important, since these proteins are misregulated in diseases like cancer. Our studies of *C. elegans* embryonic epidermal morphogenesis identified the GTPase CED-10/Rac1 as an essential activator of F-actin. However, we need to identify the GEF, or Guanine-nucleotide Exchange Factor, that activates CED-10/Rac1 during embryonic cell migrations. The two-component GEF, CED-5/CED-12, is known to activate CED-10/Rac1 to promote cell movements that result in the engulfment of dying cells during embryogenesis, and a later cell migration of the larval Distal Tip Cell. It is believed that CED-5/CED-12 powers cellular movements of corpse engulfment and DTC migration by promoting F-actin formation. Therefore, we tested if CED-5/CED-12 was involved in embryonic migrations, and got a contradictory result. CED-5/CED-12 definitely support embryonic migrations, since their loss led to embryos that died due to failed epidermal cell migrations. However, CED-5/CED-12 inhibited F-actin in the migrating epidermis, the opposite of what was expected for a CED-10 GEF. To address how CED-12/CED-5 could have two opposing effects on F-actin, during corpse engulfment and cell migration, we investigated if CED-12 harbors GAP (GTPase Activating Protein) functions. A candidate GAP region in CED-12 faces away from the CED-5 GEF catalytic region. Mutating a candidate catalytic Arginine in the CED-12 GAP region (R537A) altered the epidermal cell migration function, and not the corpse engulfment function. We interfered with GEF function by interfering with CED-5's ability to bind Rac1/CED-10. Mutating Serine-Arginine in CED-5/DOCK predicted to bind and stabilize Rac1 for catalysis, resulted in loss of both ventral enclosure and corpse engulfment. Genetic and expression studies strongly support that the GAP function likely acts on different GTPases. Thus, we propose CED-5/CED-12 support the cycling of multiple GTPases, by using distinct domains, to both promote and inhibit F-actin nucleation.

**Funding:** This research was funded by a grant from the National Institutes of Health (NIH) (GM081670) to M.C.S., and used a Spinning Disk Microscope acquired through an NIH Shared Instrumentation Grant (1S10OD010572) to M.C.S. This funding from the NIH was used for salary support for MCS, TV, SM and YP. The funders had no role in study design, data collection and analysis, decision to publish, or preparation of the manuscript.

**Competing interests:** The authors have declared that no competing interests exist.

## Author summary

GTPases in their active state promote actin nucleation that drives cellular events, from cell migrations, to cell shape changes, to cell-cell interactions. To function correctly, GTPases need to cycle from the active, GTP-bound state, to the inactive, GDP-bound state. This cycle is supported by Guanine-nucleotide Exchange Factors, or GEFs, that support activation as GDP is switched for GTP, and GTPase-Activating Proteins, or GAPs that support hydrolysis back to the GDP bound state. The Rac1/CED-10 GTPase has a well-studied GEF, CED-5/CED-12, that promotes Rac1 activation during cell engulfment of dying cells. Here we tested if CED-5/CED-12 also functioned as the activator, or GEF for Rac1 during embryonic epidermal cell migrations. Surprisingly, CED-5/CED-12 behaved completely opposite to what was expected during epidermal cell migration. Therefore, we investigated if CED-5/CED-12 could harbor a GAP function. Comparing models of human and *C. elegans* protein structures suggested a putative GAP region, which we mutated to show that CED-12 likely functions as a GAP. Genetic and gene expression tests identify other GTPases, CDC-42 and RHO-1, as likely targets of this newly uncovered CED-12 GAP function. These findings suggest CED-5/CED-12 can support the cycling of multiple GTPases, and either promote or inhibit F-actin nucleation, using its GEF or GAP functions.

## Introduction

Organized F-actin is required for cell migrations, and for engulfment of dying cells and cellular debris. The idea that the processes of cell migration and cell/corpse or debris engulfment use related actin regulators has been proposed for processes like macrophage migration and engulfment in *Drosophila* [1]. The idea that embryonic cell migrations and cell corpse engulfment are linked in *C. elegans* was suggested when we reported that loss of the Arp2/3 regulators, *gex-2* and *gex-3*, resulted in both long-lived corpses and epidermal migration defects, similar to loss of *Rac1/ced-10* [2]. These studies raised the question of how actin is regulated during two apparently distinct processes, epidermal enclosure, a collective cell migration by the epidermal cells to engulf the entire embryo, and corpse engulfment, where part of the membrane of individual cells reorganizes to engulf their dying neighbors.

Epiboly, the process that creates the metazoan body plan, depends on regulated sheet migrations. In *C. elegans*, epiboly begins at the 400-cell stage, when the main embryonic tissues begin to differentiate, and the epidermis begins its complex migrations (Fig 1A).

An open question in all organisms, is how signals are coordinated in tissues to promote the correctly oriented movements. By focusing on the first movements of the epidermal cell sheet, ventral enclosure and dorsal intercalation, we have identified a pathway that promotes polarized enrichment of branched actin: the GTPase CED-10/Rac1 activates the WAVE/Scar complex, a nucleation promoting factor (NPF) for Arp2/3. Our studies identified signals that localize CED-10 and WVE-1 correctly at membranes, to direct F-actin enrichment [3]. However, how these membrane signals are communicated to the CED-10 GTPase during epiboly is not understood.

Mutations in CED-10/Rac1 were first identified for their role in cell corpse engulfment [4]. The original *ced-10* alleles were hypomorphic alleles that interfered with the ability of cells to engulf dying cells [4,5]. Identifying null alleles of *ced-10* revealed a second role for CED-10: promoting embryonic morphogenesis [6,2]. CED-10 was proposed to recruit and activate the WAVE complex, which in *C. elegans* is essential for embryonic organ tissue formation and

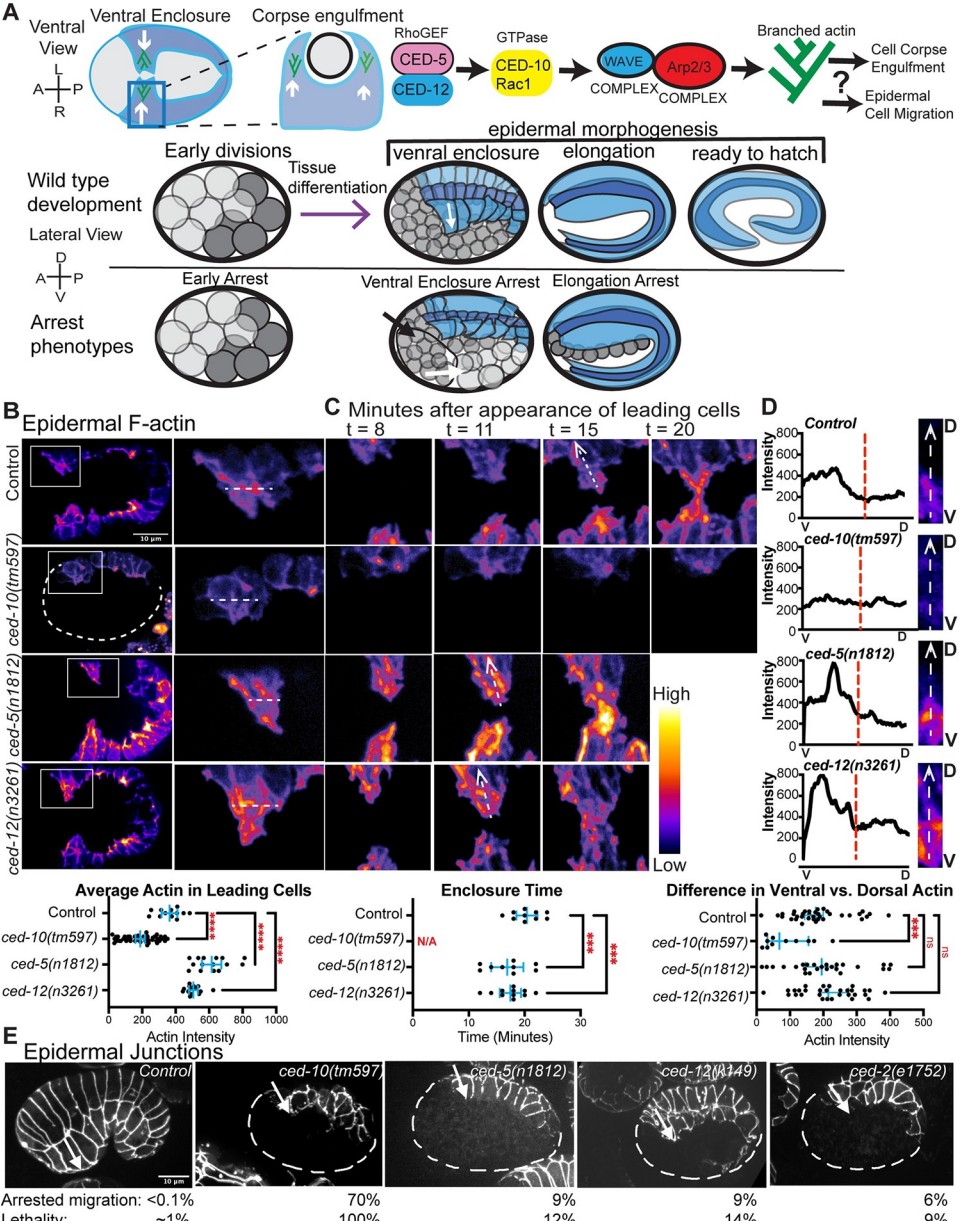

**Fig 1. CED-5/CED-12 DOCK/ELMO regulate morphogenesis.** (A). Cartoons introducing ventral enclosure, a collective epidermal cell migration essential for embryonic development. During ventral enclosure the same epidermal cells also engulf cell corpses (see also Fig 4). The proteins proposed to regulate branched actin to regulate these movements are shown: the GEF CED-5/CED-12 (DOCK/ELMO) activates CED-10/Rac1, which activates the WAVE complex, a nucleation promoting factor for Arp2/3 complex. To determine when proteins are needed during embryonic development, we measured three stages when mutant embryos died: (1) Early arrest, with no tissue differentiation, (2) Ventral Enclosure arrest, tissues differentiate but there is complete failure during the cell migration of ventral enclosure, so that internal organs like pharynx (black arrow) and intestine (white arrow) end up on the outside and (3) Elongation arrest, where the epidermis migrates to enclose the embryo, even partially, and the embryo fails the actin-based circumferential constrictions that shape the animal into a long tube [67]. (B) Epidermal F-actin in live embryos was imaged using the *lin-26p::LifeAct::mCherry* strain [20]. Embryos were oriented ventral up and anterior to the left. Two Leading cells (LCs) on one side were magnified to the right (white box). White dotted line across the brightest region of the LCs shows how actin intensity was measured, and added to the plot below. A dotted line for *ced-10(tm597)* indicates the rest of the embryo, which is not enclosed, and only one side of epidermis is visible. Each dot in the graph represents the average of three measurements. n = at least 12 embryos. (C) Comparison of the time it takes for the LCs to meet at the midline, showing selected time points after their first appearance (t = 8, 11, 15, and 20 minutes). Enclosure time, plotted below, was measured from the first appearance of LCs. *ced-10(tm597)* was

marked as N/A as LCs never meet. (D) Representative line scans from ventral (V) to dorsal (D) illustrate distribution of actin intensity from the ventral half compared with the dorsal half of LCs. Close-ups of representative Leading Cells, ventral (bottom) to dorsal (top). A ventral to dorsal line was drawn, as shown in the t15 Control, dotted white arrow, and intensity was plotted using Fiji's Plot Profile tool. Dashed red lines indicate half the cell's length. Graph illustrates the difference between average F-actin in ventral vs. dorsal half of the leading cells, measured with the freehand tool in ImageJ. (E) Morphogenesis defects visualized by the *dlg-1::gfp* (FT48, [23]*)* transgene that marks the junctions of epithelial tissues. Embryos are age matched, ~360 min. at 23˚C, so that the Control embryo has enclosed, while equally aged mutants have arrested. Since embryos are highly curved, each image is a Z projection of 5 slices made 0.5 microns apart. White arrows point to same row of ventral epidermal cells in the wild type and mutant embryos. The % of embryos showing arrested migration, and total lethality, is indicated below. See also Table 1. In this and all Figures error bars (blue) show 95% confidence intervals. Asterisks mark statistical significance, * = p<0.05, ** = p<0.01, *** = p<0.001, **** = p<0.0001, as determined by one-way Welch's ANOVA test.

tissue migrations [2,7]. GTPases are proposed to be recruited and activated at membranes by guanine-nucleotide exchange factors, GEFs, that enhance the exchange of GDP for GTP, thus activating the GTPase (Reviewed in [8,9]). The GEF for CED-10 during corpse engulfment was proposed to be CED-5/DOCK180 [5] CED-5 was also proposed to act as the GEF for CED-10 during neuronal migrations [10]. However, the CED-10 GEF or GEFs that promote embryonic epidermal movements are not known.

CED-5/DOCK180 is a DOCK GEF which requires an ELMO protein to activate Rac [11]. In *C. elegans*, CED-5/DOCK180 works with CED-12/ELMO to activate CED-10 during corpse clearance (Proposed by [12]; [13–17]). During corpse engulfment CED-5/CED-12 and CED-10 are proposed to act in neighboring cells, to enclose dying cells, by promoting F-actin polymerization in membrane protrusions that encircle dying cells. F-actin was shown to enrich around dying cells in the germline and embryos [18,19], and then disappear as the corpse decayed. While some studies noted that CED-12 appears to have an embryonic role [13], the role of CED-5 and CED-12 during embryonic development, and epidermal morphogenesis has not been investigated.

We present here analysis of the role of CED-5/DOCK and CED-12/ELMO, during epidermal morphogenesis, that shows these cell death regulators also support epidermal migrations. We therefore investigated if CED-5/CED-12, a candidate GEF for CED-10's ventral enclosure function, shares epidermal *ced-10* phenotypes. While loss of the candidate GEF, CED-5/CED-12 reduced F-actin around corpses, as expected, loss of CED-5/CED-12 increased F-actin in epidermal cells during ventral enclosure, the opposite phenotype as loss of CED-10. To determine how a GEF can both promote and inhibit F-actin formation, we investigated if this candidate CED-10/Rac1 GEF also functions as a RhoGAP. A candidate GAP domain was identified in CED-12 and mutated. Loss of the GAP function of CED-12 resembled loss of CED-12 for epidermal migrations, but corpse engulfment was normal. These studies identify a previously undescribed role during embryonic morphogenesis for CED-5 and CED-12, proteins well-studied for their cell corpse engulfment. Analysis of a mutation in the newly identified GAP domain of CED-12 suggest CED-12/ELMO, uses different subdomains to switch from promoting to inhibiting F-actin formation, depending on the subcellular context. Two likely GTPase targets of the CED-5/CED-12 epidermal morphogenesis function are identified, placing DOCK/ELMO at a central position for actin regulation during morphogenesis.

## Results

### CED-5/DOCK and CED-12/ELMO regulate morphogenesis by modulating F-actin in migrating epidermal cells

CED-5/DOCK-180 and CED-12/ELMO act as a bipartite GEF that promotes active CED-10/Rac1 during the engulfment of cells that die by programmed cell death in *C. elegans* [13–17].

We showed that CED-10/Rac1 also acts to promote the tissue migration known as ventral enclosure that leads the epidermis to enclose the developing *C. elegans* embryo [2](Fig 1A). However, the GEF for CED-10/Rac1 during this process is not known.

To test if CED-5 and CED-12 are required during embryonic epidermal cell migrations, we crossed null or strong loss of function alleles of *ced-5(n1812)* [20] and *ced-12(n3261)* [17] into a strain that expresses F-actin only in the epidermis (*lin-26p*:: *Lifeact*::*mCherry*, Fig 1B, strains are listed in the Methods)[21]. Staged embryos that were oriented with ventral tissues facing up (towards the cover slip) were imaged from 280 to 320 minutes after first cleavage. These 4D movies of live embryos showed that wild type embryos are enriched for F-actin at the leading edge of the migrating row of ventral cells, as previously shown [3,22]. Timing the migration, from the first appearance of leading cell edge protrusions, to when the leading cells met at the ventral midline, took approximately 20 minutes at 23˚C (Fig 1C). We use the term "protrusion" here and throughout, since the leading cells (and all cells along the ventral edge) make membrane extensions that encompass both lamellipodia and filopodia (see also Fig 2C). Loss of the GTPase CED-10/Rac1, showed significantly reduced levels of F-actin in leading cells, and arrested migrations, as would be expected for the removal of a major activator of branched actin (Fig 1B–1D). Only one leading cell is visible in the *ced-10(tm597)* embryo due to the enclosure failure and the angle of the movie. If CED-5/CED-12 are the GEF that activates CED-10 during ventral enclosure, we expected removing them would similarly lead to decreased F-actin levels and slower or arrested migrations. Surprisingly, animals carrying null mutations in *ced-5* or *ced-12* displayed two opposite phenotypes to loss of CED-10. First, the levels of F-actin in the leading cells were significantly higher than in controls (Fig 1B). Second, the ventral cells met at the midline significantly faster, on average 5 minutes, or 25% faster (Fig 1C). We obtained the same results with multiple null or strong loss of function alleles, including *ced-5(n1812)*, *ced-12(n3261, ky149)*, and when *ced-5* was removed by RNAi. Therefore, the candidate GEF for CED-10 during ventral enclosure had the opposite effect on F-actin levels and migration timing as was expected.

To test if branched actin regulators were also affected, we measured the effect of *ced-5 (n1812)* on the rescuing transgene *ced-10*::*gfp* or endogenously tagged *gfp*::*wve-1 (OX466; OX699)*[3,23]. Loss of CED-5 resulted in significantly increased expression of *ced-10*::*gfp* and *gfp*::*wve-1* in many tissues, including in the epidermal cells at the time of their migration (Fig 2A and 2B). Thus, loss of *ced-5* led to increased overall F-actin in the migrating epidermal cells (Fig 1B) that correlated with increased branched actin regulators (Fig 2A and 2B).

Proper migration of the embryonic epidermis requires dynamic F-actin at the leading edge, that is ventrally enriched [3]. The *ced-5* and *ced-12* mutants showed properly polarized, ventrally enriched F-actin, as shown by line scans and comparison of average ventral and dorsal F-actin (Fig 1D). To test if the higher actin levels (Fig 1B), and faster migrations (Fig 1C) correlated with increased dynamics at the leading edge of migrating cells, we measured protrusions and retractions. While a null mutation in *ced-10*, *tm597*, led to strongly decreased protrusions and retractions, null mutations in *ced-5* and *ced-12* resulted in significantly increased protrusions, with a more modest effect on retractions (Fig 2C). Elevated F-actin and more rapidly moving cells may disrupt the migrations. If this was the case, we expected to see altered behavior in the migrating epidermal cells.

## Changes in F-actin levels and cell migration speed contribute to morphogenesis defects

Since loss of CED-10, and of its proposed GEF CED-5/CED-12 caused opposite effects on epidermal F-actin levels, we next examined the consequences of the mutations for embryonic

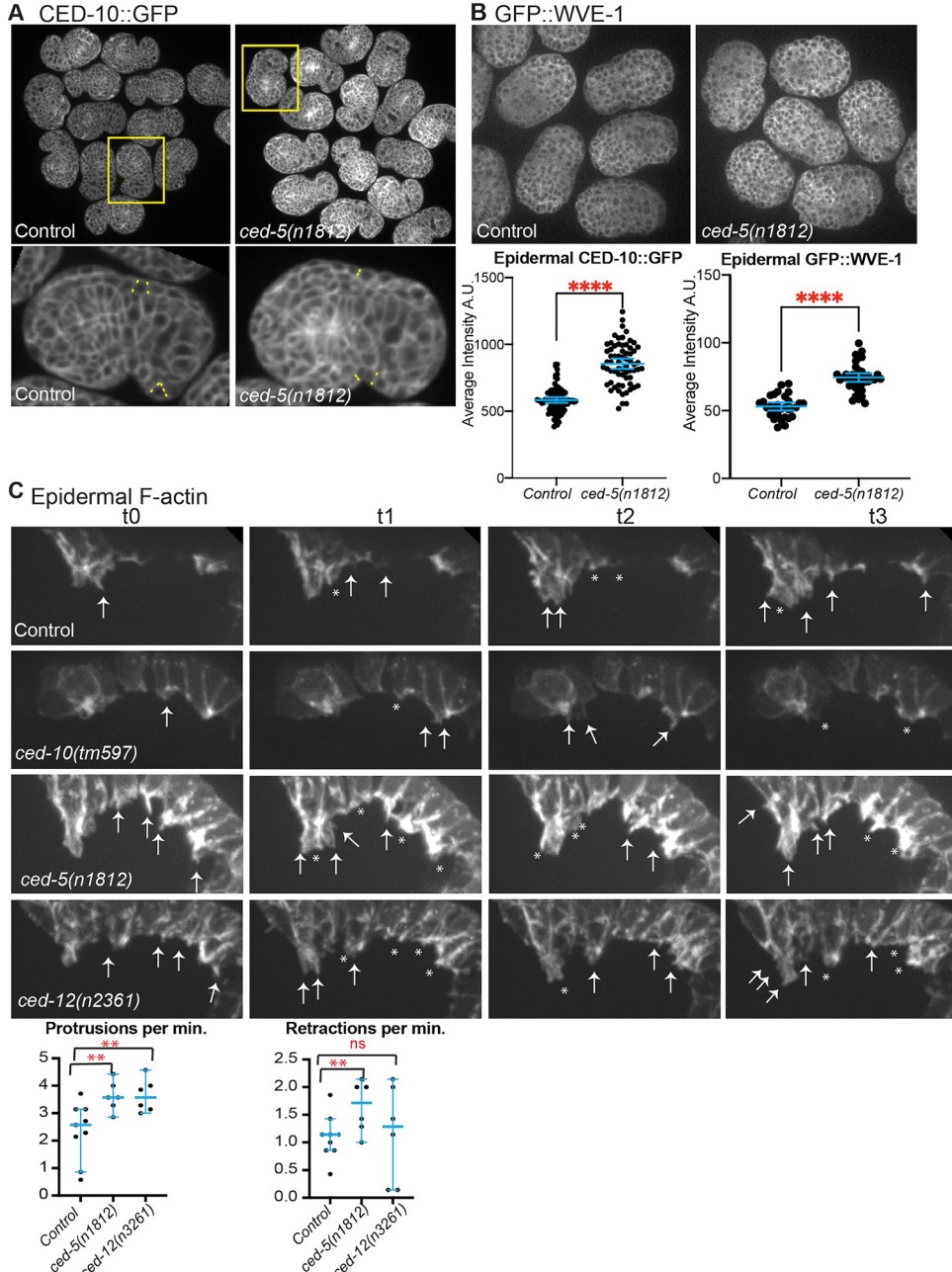

**Fig 2. CED-5/CED-12 DOCK/ELMO regulate levels of branched actin regulators and cell dynamics.** (A). Staged movies of control *ced-10::gfp* embryos (*OX466* [3]) and equally staged *ced-10::gfp; ced-5(n1812)* embryos were compared for average levels of *ced-10::gfp* at lateral regions of epidermal cells at early bean stage. Dotted yellow lines in the enlarged embryos show regions that were measured, at the lateral boundaries of epidermal cells. (B) Staged movies of control *gfp::wve-1* embryos (*OX699* [22]) and equally staged *gfp::wve-1; ced-5(n1812)* embryos were compared for average levels of epidermal *gfp::wve-1*, at the early bean stage, and measured as in (A). (C) Movies using the epidermal F-actin strain *lin-26p::LifeAct::mCherry* 1 were enlarged, on just the left side, during ventral enclosure, to illustrate the protrusions (arrows) and retractions (asterisks) continuously happening along the edge of the migrating epidermal cells. While we measured dynamics only from the two Leading Cells on each side for the graphs, there are protrusions all along these cells. Graphs: Protrusions and retractions of at least 6 sets of Leading Cells per genotype were counted during enclosure, from LCs' first appearance, for 7 time points, and plotted as protrusions or retractions per minute. Images are shown at the same settings. Asterisks mark statistical significance, * = p<0.05, ** = p<0.01, *** = p<0.001, **** = p<0.0001, as determined by one-way Welch's ANOVA test.

morphogenesis (Fig 1A). When regulators of branched actin are removed, most embryos die due to cell migration arrests at two stages. Most embryos fully differentiate tissues, but without CED-10/Rac1, or the WAVE complex, or the Arp2/3 complex, most embryos fail ventral enclosure due to complete lack of epidermal migrations (Fig 1A)[2,7]. Partial loss of these branched actin regulators leads to partial enclosure, and epidermal elongation arrest (Fig 1A). These arrests can be monitored with differential interference contrast (DIC) optics, and by following the epithelial apical junctions using transgenic strains like *dlg-1::gfp* (FT48)[24] (Fig 1E). We compared the frequency of embryonic lethality with a morphogenesis phenotype, for embryos missing *ced-10*, *ced-5*, *ced-12* or the adaptor *ced-2/CRKII*, which is thought to act with *ced-5* and *ced-12* to support corpse engulfment [24,4,5,15,17]. Complete loss of *ced-10* is as severe as complete loss of the WAVE complex, with 100% of the embryos dying with a Gex (gut on the exterior) Ventral Enclosure arrest phenotype [7]. Partial reduction mutations in *ced-10*, like *n1993*, a mutation in the CED-10 prenylation site, thought to reduce membrane attachment [5], results in partially penetrant lethality, of approximately 11%, with Ventral Enclosure and Elongation arrests [7]. Loss of *ced-5* or *ced-12* using putative null alleles led to partially penetrant lethality (9–16% lethality), with the majority of the deaths due to ventral enclosure arrest, similar to the partial loss of function *ced-10* alleles (Fig 1E and Table 1). Mutations in *ced-2* had a similar phenotype (9% lethality, 5% due to ventral enclosure arrest, Fig 1E and Table 1). Thus, 100% of the *ced-5* and *ced-12* mutant embryos had elevated F-actin, which led to partial arrest, during epidermal morphogenesis. These results reinforced that

**Table 1. Embryonic lethality count of *ced-5*, *ced-12* and *ced-2* alleles.**

| Genotype | n | Lethal | %Arrest Stage | | | Total % Lethal | Signif. |
| --- | --- | --- | --- | --- | --- | --- | --- |
| | | | Early | VE | El | | |
| N2, *Wild-type* Control | 1610 | 18 | 0.25 | 0.3 | 0.6 | 1.1 | |
| *ced-5(n1812)* | 713 | 83 | 2.2 | 8.3 | 1.1 | 12 | **** |
| *ced-12(k149)* | 527 | 72 | 4.6 | 7.4 | 1.7 | 14 | **** |
| *ced-12(n3261)* | 340 | 49 | 0.8 | 8 | 5.6 | 14 | **** |
| *ced-2(e1752)* | 500 | 46 | 1.5 | 5 | 2.5 | 9 | **** |
| *ced-12(pj74) R537A-GAP* | 581 | 40 | ND | ND | ND | 7 | **** |
| *ced-12(pj75)-PXXXP-AAA* | 470 | 18 | ND | ND | ND | 4 | **** |
| *ced-12(k149); ced-2(e1752)* | 1167 | 105 | 2.3 | 5.5 | 1 | 9 | ns vs. singles |
| *ced-5(pj81) SR1541/1542AA RBD* | 1102 | 249 | 1.4 | 16 | 8.25 | 23 | **** |
| L4440 Control RNAi | 653 | 5 | 0. 5 | 0.15 | 0.15 | 0.77 | |
| *ced-5 RNAi* | 539 | 55 | 3.9 | 4.3 | 2 | 10 | |
| *ced-12(k149); L4440 RNAi* | 735 | 64 | 2.7 | 4.6 | 1.4 | 9 | |
| *ced-12(k149); ced-5 RNAi* | 551 | 60 | 3.4 | 6.2 | 1.3 | 11 | ns vs. singles |
| *ced-2(e1752); L4440 RNAi* | 763 | 57 | 2.2 | 4.3 | 0.9 | 7 | |
| *ced-2(e1752); ced-5 RNAi* | 726 | 76 | 3.7 | 5.4 | 1.4 | 10 | ns vs. singles |
| *ced-12(k149); ced-2(e1752); L4440 RNAi* | 417 | 47 | 4.3 | 6.2 | 0.7 | 11 | ns vs. singles |
| *ced-12(k149); ced-2(e1752); ced-5 RNAi* | 540 | 63 | 2.6 | 7.6 | 0.9 | 12 | ns vs. singles |

Right column: Statistical significance relative to N2 Controls, or single mutants, top section. Bottom section: compared to *ced-5 RNAi*, L4440 empty vector RNAi control, or the single mutants on control RNAi. Asterisks mark statistical significance

\* = p<0.05

\*\* = p<0.01

\*\*\* = p<0.001

\*\*\*\* = p<0.0001

ns = not significant. Student two-tailed T test.

high levels of F-actin can interfere with morphogenesis, as we previously showed for loss of *vab-1* [3], *hum-7* [25] and *rga-8* [21].

## Embryo shape changes support that CED-5 and CED-12 function in early embryos

Since the embryonic role of CED-5 and CED-12 has not been investigated, we looked for further evidence that CED-5 and CED-12 are required in embryos. We noted that some embryos that die due to arrested epidermal cell migrations also displayed a round shape instead of the wild type oval shape (Fig 3A). Embryo shape is established just after fertilization: unfertilized oocytes are round, and become oval after fertilization, as sperm entry is coupled to release of chitin and other egg shell components [26–30]. Mutations that alter constriction of the spermatheca during fertilization result in round or otherwise misshapen embryos [31,32].

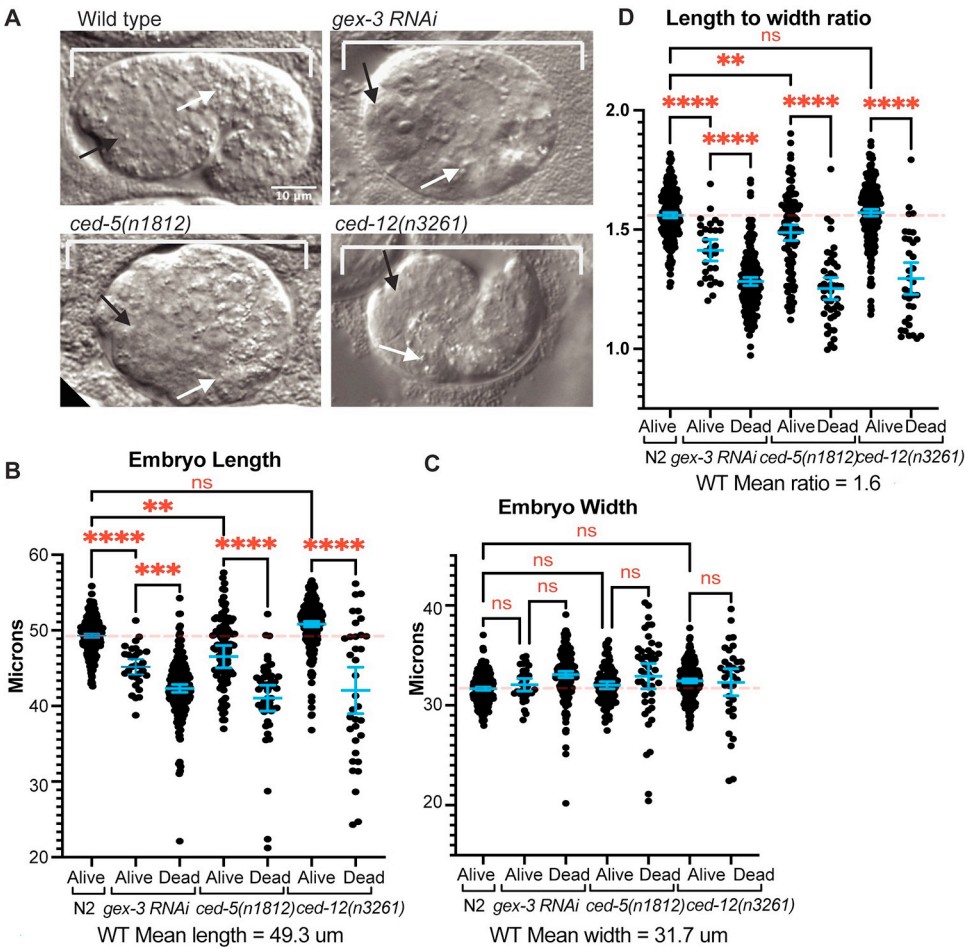

**Fig 3. Mutant embryos that die are shorter and rounder than those that live.** A. Representative differential interference contrast (DIC) images of embryos, oriented anterior to the left, and dorsal up, to illustrate a live Control embryo (Wild Type) and dying, short round embryos with reduced *gex-3*, *ced-5* or *ced-12*. Black arrow: anterior pharynx; white arrow: anterior intestine, both of which end up on the outside of embryos that fail ventral enclosure. B. Embryo length was measured in Live and Dead embryos for each genotype. C. Embryo width was measured in Live and Dead embryos for each genotype. D. The ratio of length divided by width was plotted for each genotype. At least 100 embryos were measured for each genotype. Asterisks mark statistical significance, * = p<0.05, ** = p<0.01, *** = p<0.001, **** = p<0.0001, as determined by one-way Welch's ANOVA test.

Mutations in WAVE complex components like *gex-3*, or null alleles of *ced-10* are frequently round [2,7]. To compare the "round" vs. oval wild-type phenotypes, we measured the lengths and widths of embryos, and documented if the embryo lived or died. In Wild Type, the average length was 49.3um and the average width was 31.7um. Comparing wild type with *ced-5 (n1812)*, *ced-12(n3261)* and *gex-3(RNAi)* showed significantly reduced lengths in the majority of mutant embryos, with the exception of *ced-12(n3261)* viable embryos (Fig 3A and 3B). By comparison, though the widths were slightly increased in some mutants, the differences were not significant (Fig 3C), suggesting the main differences in mutant embryos are in their lengths. In wild-type embryos, the ratio of length divided by width averaged 1.6, while mutations in *ced-5 (n1812)*, *ced-12(n3261)* and *gex-3(RNAi)* resulted in significantly reduced ratios for all *ced-5(n1812)* and *gex-3 RNAi* embryos, and for *ced-12(n3261)* embryos that died (Fig 3D).

Since the loss of *ced-5 (n1812)*, *ced-12(n3261)* and *gex-3(RNAi)* resulted in a mixture of live and dead embryos, we compared how length correlated with viability. For all three mutations, dead embryos were significantly more likely to be "round" or short. In Wild Type, <1% of embryos are "round" or short, and <1% of embryos die. *ced-5 (n1812)* and *gex-3(RNAi)* embryos that lived were significantly shorter compared to wild type. However, *ced-5 (n1812)*, *ced-12(n3261)* and *gex-3(RNAi)* embryos that died were much shorter than wild type, and significantly shorter than their sibling embryos that lived. The ratio of length divided by width dropped from 1.6 in wild type to ~1.3 in dying mutants (Fig 3D). Altogether these embryo shape changes suggested loss of *gex-3*, *ced-5* or *ced-12* affected early events of embryonic development, when embryo shape is first established.

## CED-5/DOCK, CED-12/ELMO and CED-2/CRKII together support embryonic viability

The CRKII homolog, CED-2, is part of the corpse engulfment pathway that includes CED-5 and CED-12. To test if *ced-5*, *ced-12* or *ced-2* work in a complex during ventral enclosure as they do during corpse engulfment, we constructed double mutants between them. All double mutants had similar levels of embryonic lethality as the single mutants, with most deaths due to ventral enclosure arrests. To test the triple mutant, we depleted *ced-5* via RNAi in animals with the *ced-12; ced-2* double mutation (Table 1). These animals had similar levels of lethality, with Gex-like morphogenesis phenotypes supporting that the three proteins also act together during morphogenesis (Table 1).

We used a genetic test to measure if *ced-5* behaved like a *ced-10* GEF. We predicted that loss of a GEF would strongly enhance partial loss of a GTPase. Combining two different hypomorphic alleles of the GTPase *ced-10 (n1993 and n3246)* with depletion of *ced-5* via RNAi resulted in no significant change in embryonic lethality (Table 2). Thus, CED-5 did not behave like a candidate GEF for CED-10/Rac1 ventral enclosure function.

**Table 2. Effect of *ced-5* null alleles on partial loss of function *ced-10* alleles.**

| Genotype | n | %Lethality | Signif. |
|---|---|---|---|
| *ced-10(n1993)–L4440 RNAi control* | 360 | 11 | |
| *ced-10(n3246)–L4440 RNAi control* | 448 | 12 | |
| *ced-5 RNAi* | 414 | 12 | |
| *ced-10(n1993); ced-5 RNAi* | 357 | 14 | ns vs. singles |
| *ced-10(n3246); ced-5 RNAi* | 286 | 13 | ns vs. singles |

## CED-5/CED-12 and GEX-3 promote F-actin at corpses

To test if CED-10, and its proposed GEF CED-5/CED-12, have similar or opposite effects on F-actin around corpses, we crossed putative null mutations in *ced-5* or *ced-12* into a strain with labeled epidermal F-actin (*lin-26p::LA::mCherry*, used in Fig 1) and which also expressed the receptor, CED-1, around corpses (*Pced-1::ced-1::gfp*)[33]. These movies were used to measure F-actin levels around well-studied corpses, C1 (left), C2 (right) and C3 (left) in the ventral epidermis [18]that are engulfed by the same migrating epidermal cells (Fig 4A and 4B). Loss of *gex-3*, an effector of CED-10 with cell migration and corpse engulfment defects [2], resulted in reduced F-actin levels around the corpses, as did loss of *ced-5(n1812)*, or *ced-12(n3261)*. Since all of the mutants have long-lived corpses that last longer than in wild type controls, this result agrees with the model that F-actin enrichment around corpses is promoted by a molecular pathway that includes the GTPase CED-10, its effector the WAVE complex, and its GEF CED-5/CED-12, to engulf and remove corpses during development [34,17] (Fig 4C). Thus, monitoring F-actin in embryonic epidermal cells showed two opposing results: during corpse engulfment, loss of CED-5/CED-12 caused the expected loss of F-actin enrichment around corpses (Fig 4), while during ventral enclosure (Figs 1 and 2), loss of CED-5/CED-12 caused the opposite effect, increased F-actin (Fig 1B), and branched actin regulators (Fig 2A and 2B). These opposing effects on F-actin occur in the same tissue (Fig 4D).

## CED-12 Alignments and structural comparison identify candidate GAP motif

To address how one complex, the CED-5/CED-12 GEF, could promote two distinct and opposite functions, we looked for evidence of alternative forms of CED-5 or CED-12. The current molecular model for CED-5 predicts a single protein, and the model for CED-12 predicts two proteins that differ only by 7 amino acids at the N terminus [35]. While CED-12 is the only ELMO protein in *C. elegans*, other organisms, including humans and Dictyostelium, have six ELMOs, including some thought to promote F-actin formation, and others thought to inhibit F-actin. Inhibitory ELMOs, like ELMO-A in Dictyostelium and ELMOD1 in humans may use a GAP domain [36]. Alignments of CED-12 with *C. elegans* GAP proteins revealed a region with high homology to GAPs (Figs 5A and 5B and S1A). Aligning CED-12 with ELMOs and ELMODs showed that CED-12, which is similar in length to the ELMOs, aligns all along its length with the ELMOs (Fig 5A). The shorter ELMODs have a conserved GAP domain [37] that CED-12 does not share. Instead, CED-12's GAP domain is in a region only found in the longer ELMO proteins. While CED-12 has the necessary Arginine, R537, required for GAP function, human ELMOs have similar surrounding residues but substituted R with Q or Ks (Figs 5A and 5B, and S1A). Thus, the remnants of a GAP domain may remain in the ELMOs. To determine the location of the proposed GAP region, we used the structures.

We aligned, or threaded, the recently published structure of DOCK2/ELMO1 with active Rac [38],with *C. elegans* CED-5/CED-12 with active CED-10 using the program SWISS-MODEL and visualized them using UCSF Chimera [39–41] (Fig 5B). We also aligned the closed structure [38] of DOCK2/ELMO1 with CED-5/CED-12 (Fig 5D). The aligned structures showed that the proposed CED-12 GAP's catalytic Arginine, R537, faces away from the membrane, and away from DOCK/CED-5 and Rac1/CED-10. This is true for CED-12 R537 aligned to the open and closed conformations of DOCK2/ELMO1 (Fig 5B–5D). Thus *C. elegans* CED-12 can be fit onto the structure of the human ELMO, yet it contains a catalytic Arginine, not found in this region of human ELMOs, that points away from the pocket of CED-5 that binds to CED-10. For example, human ELMO1 replaces R527 with Q541, but the structures

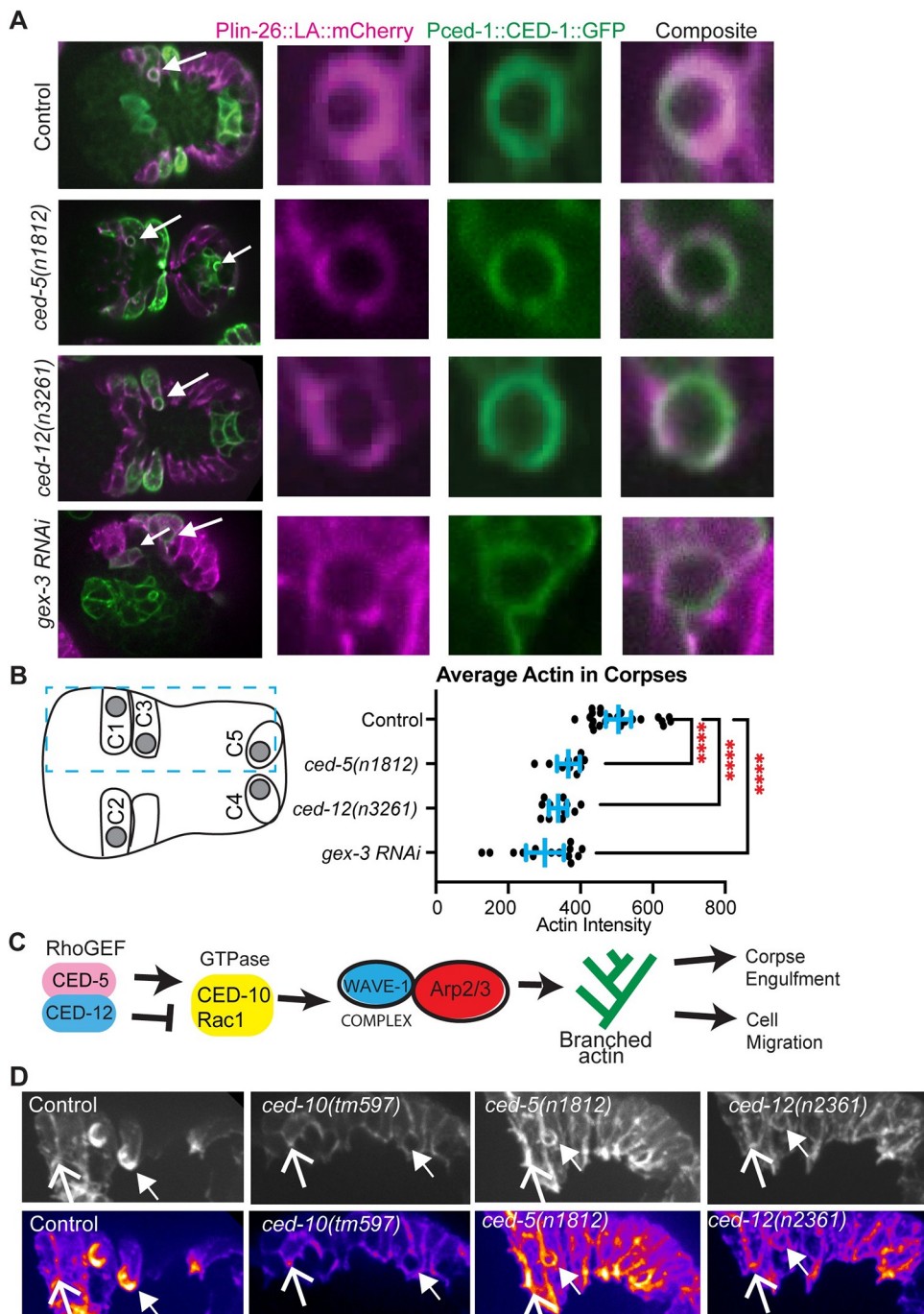

**Fig 4. CED-5 and CED-12 promote F-actin around corpses.** (A) To visualize corpse engulfment by epidermal cells, embryos with epidermal F-actin (*lin-26p::LifeAct::mCherry*), and with the corpse receptor (*ced-1p::ced-1::gf*) were imaged live, at the same times in development. Left panels: composite image of the full embryo, larger white arrows indicate C1 corpse, smaller arrows indicate other corpses. The right panels zoom in on C1 corpse, F-actin in magenta, CED-1::GFP in green. (B) The average intensity of F-actin around C1, C2, and C3 corpses, labeled in the cartoon, was measured, n = at least 20 cells. By using the line scan function of ImageJ, 2 measurements were made at the anterior and posterior sides of the corpse, averaged across 3 different z-planes for each side, at 3, 4, and 5 minutes after the first appearance of actin around the corpse (~330 minutes of development). Asterisks mark statistical significance, **** = p<0.0001, as determined by one-way Welch's ANOVA test. (C) CED-5/CED-12 are proposed to have two different roles regulating F-actin during two branched-actin dependent events in the same cells. (D) To illustrate the contrasting roles, staged embryos with epidermal F-actin (*lin-26p::LifeAct::mCherry*) are shown at similar stages, when Controls show much higher levels of F-actin around C1 and C3 corpses (closed arrow), relative to epidermal cell boundaries

(open arrow). The bottom row is the images from the top row, in ImageJ Fire setting. Dotted blue box in (B) shows region enlarged in (D).

otherwise align (Fig 5A and 5B). Since R537 is not buried in a pocket in the open or closed conformation, it may be open to interactions with other proteins.

## CED-12 GAP (R537A) and CED-5 Rac1 Binding (SR1541/1542AA) mutations affect embryonic development

If the predicted GAP motif detected in CED-12 is important for function, mutating it may indicate how this proposed GAP function supports CED-12 activities (Fig 5). We predicted that mutating the GAP function would be different from completely deleting *ced-12*, since the rest of CED-12 would be available to support the CED-5/CED-12 GEF function.

GAP function requires a catalytic Arginine [42,43] We used CRISPR to mutate the R537 to Alanine, predicted to eliminate catalytic GAP function [25]. We first examined the *ced-12 (pj74)* R537A allele for embryonic lethality. We found that 7.6% of embryos died, with mostly morphogenesis phenotypes, compared to 9–16% for two putative null alleles of *ced-12* (Table 1 and Fig 6A).

To interfere with the GEF function of CED-5/CED-12, we used the structure alignment based on DOCK2/ELMO1 and consulted the DOCK5/ELMO1 structure [44]. DOCK2 has a helix predicted to make multiple interactions with Rac1, required for promoting its activation, and sometimes referred to as the Rac binding domain (RBD). CED-5 has a similar helix, with conserved amino acids, like DOCK2 S1528, conserved in CED-5 as S1541 (Figs 5C, S1B and S1C). M1529 of DOCK2 is replaced in CED-5 by R1542 (Figs 5C, S1B and S1C). Using UCSF Chimera [41,45], we predicted that R1542 would make similar packing interactions with Rac W56/CED-10 W56 (Figs 5B and S1C). Therefore, to interfere with the GEF activity, we used CRISPR to mutate S1541 and R1542 to Alanines. This mutant, *ced-5(p81)* SR1541/1542AA, had slightly higher embryonic lethality (23%) than the *ced-5(n1812)* or *ced-12(n3261, ky149)* null alleles, and showed similar phenotypes including unengulfed corpses (100% penetrant), and failed morphogenesis (16% ventral enclosure arrest, 8.25% elongation arrest, Table 1). We refer to this mutant as the *ced-5(pj81)* Rac1/CED-10 Binding mutant.

## CED-12 GAP (R537A) allele affects epidermal migrations, not corpse engulfment

To test if the embryonic lethality in the *ced-12(pj74)* R537A proposed GAP mutant correlated with changes in F-actin at the leading edge of migrating epidermal cells, we crossed this mutant into the epidermal F-actin strain (*lin-26p::LA::mCherry*), and found that F-actin levels increased, similarly to null alleles in *ced-12* (Fig 6A and 6B). Examining the corpses demonstrated that *ced-12(pj74)* R537A had normal corpse engulfment, and wild-type levels of F-actin around corpses, in contrast to the abnormal corpse engulfment and low levels of F-actin around corpses seen in the *ced-12(n3261)* null allele (Fig 6A, 6C and 6D). These results suggested that CED-12 has a GAP function that is required for epidermal cell migrations of ventral enclosure, but dispensable for corpse engulfment (Fig 6D).

## CED-5 (SR1541-1542AA), a CED-10/Rac1 Binding mutation, affects corpses, and epidermal migrations

The *ced-5(p81)* mutation, a two amino acid change in the Rac1 binding domain, predicted to interfere with GEF activity (Fig 5B and 5C), affected corpse engulfment, with persistent

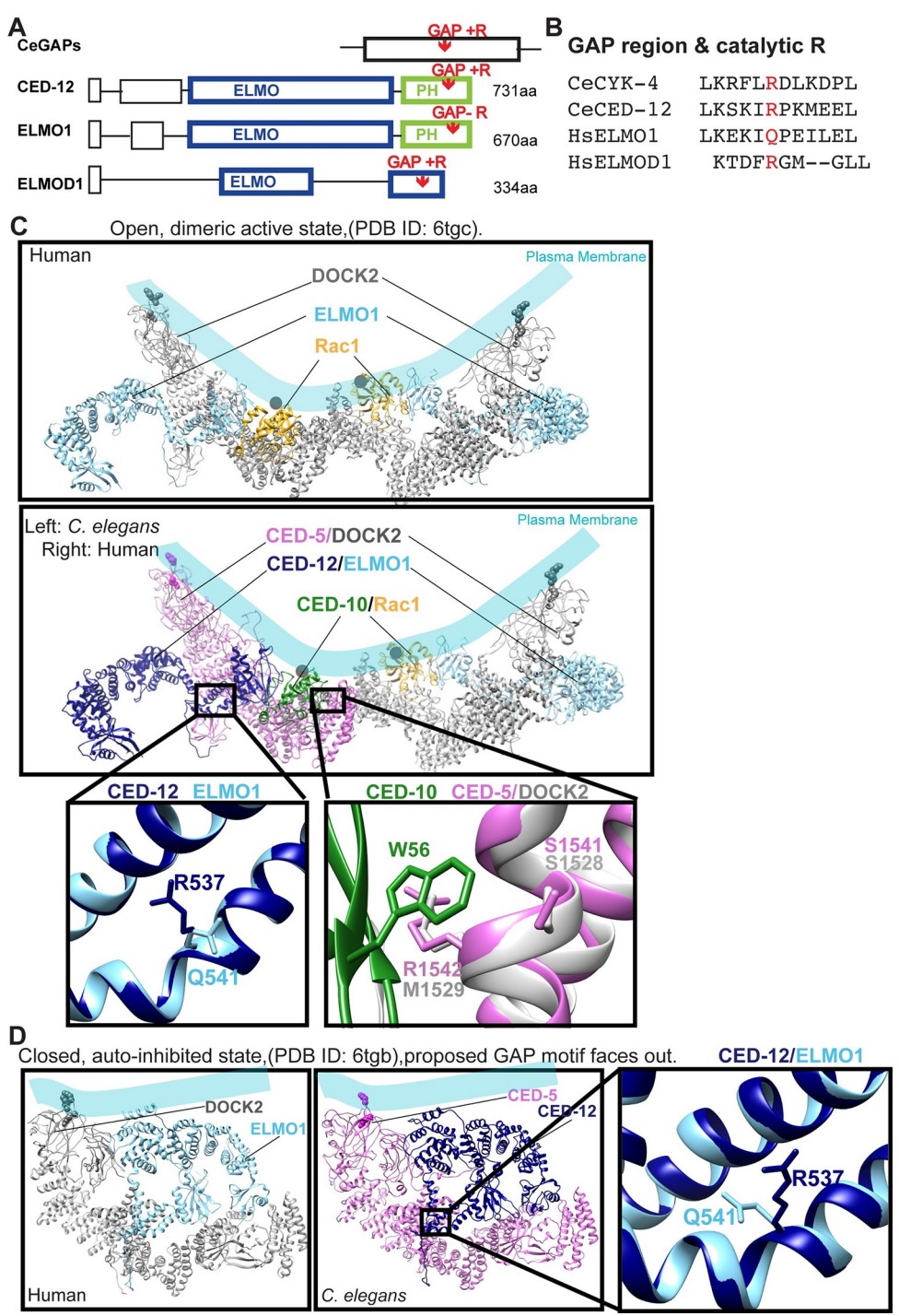

**Fig 5. Protein alignments and models identify residues for proposed GAP and GEF functions of CED-5/CED-12.**
(A) Comparison of the overall structure of CED-12 aligned to *C. elegans* GAP proteins and to human ELMO1 and ELMOD1. (B). Aligning CED-12 to the *C. elegans* GAPs like CYK-4 identified a candidate GAP region in CED-12. Aligning CED-12 to human ELMOs and ELMODs suggested all have GAP-like regions, but only CED-12 and ELMODs, not ELMOs, have the catalytic Arginine (R) required for GAP function. See also further alignments S1 Fig. (C) Ribbon model of open, dimeric active conformation of DOCK2/ELMO1/Rac1 and CED-5/CED-12/CED-10 (PDB ID: 6TGC) [38]. The top panel depicts the human structure. The panel below shows the threaded structures for *C. elegans* CED-5/CED-12/CED-10 on the left dimerized with the human homologs on the right. Pink/gray: CED-5/ DOCK2, Navy blue/light blue: CED-12/ ELMO1, green/yellow: CED-10/ Rac1. Left box: R537 in *C. elegans*, replaces Q541 in the human homolog. Right box: CED-5 aligned to the Rac1-binding domain (RBD) of human DOCK2. In the

Rac1-binding domain (RBD), CED-5 S1541/R1542 align closely to DOCK2 S1528/M1529, forming a pocket for CED-10 W56/ Rac1 W56. (D) Closed, auto-inhibited conformation of DOCK2/ELMO1 (left) and threaded models of *C. elegans* CED-5 and CED-12 on right (PDB ID: 6TGB). Boxed and magnified region shows CED-12 R537 superimposed on human ELMO1 Q541, is present on the outside of the structure. See also S1 Fig.

unengulfed corpses, and highly reduced F-actin around corpses (Fig 6A and 6C), as would be expected if this mutation altered the GEF function of CED-5/CED-12. In the migrating epidermal cells, *ced-5(p81)* significantly reduced F-actin, a distinct phenotype compared to complete loss of CED-5 (Fig 1B), or loss of CED-12 *(n3261)* or the CED-12 GAP mutant *(pj74)*, which all increase epidermal F-actin (Fig 6A, 6B and 6D). Therefore, while mutating the CED-12 GAP function only affected the cell migration function of CED-5/CED-12, and not the corpse engulfment function, mutating the Rac1/CED-10 binding ability of CED-5 affected both processes. Since the Rac1 binding mutant interferes with more than just GEF function, we focused on how the CED-12 GAP mutant, that only interferes with cell migration, and not corpse engulfment, interacted with the three GTPases of the Rho/Rac1/Cdc42 family.

## Evidence that CED-5/CED-12 regulate two other GTPases, RHO-1/RhoA and CDC-42

The CED-12 GAP motif, when threaded with the structure of DOCK2/ELMO1 bound to active Rac1 [38], is oriented away from the Rac1 binding interface (Fig 5C). This suggested that the CED-12/CED-5 GAP activity may have other GTPase targets other than Rac/CED-10 (Fig 7A). A genetic test used in *C. elegans* to identify GTPase targets is to cross the candidate GAP to hypomorphic alleles, or partial loss of function, of the target GTPases. The prediction is that loss of a GAP in combination with a hypomorphic allele of the GTPase will rescue the loss of function phenotype [46]. In contrast, we predicted that loss of a GEF in combination with a hypomorphic allele of the GTPase will synergistically enhance the loss of function phenotype. Therefore, we combined partial loss of function mutations for the three main *C. elegans* GTPases, *ced-10/Rac*, *cdc-42* and *rho-1*, with either loss of *ced-5 (n1812)*, loss of *ced-12 (n3261)*, or the CRISPR mutations generated based on the structure, *ced-12* GAP *(pj74)*.

To test for interactions with RHO-1/RhoA, we used a temperature sensitive mutation in Rho Kinase, *let-502(sb1118ts)* [47]. At 25°C *let-502(sb1008ts)* resulted in 20% embryonic lethality due to elongation defects (Fig 1A), and in combination with null alleles of *ced-5 or ced-12*, this rose to 33% and 37%, respectively, while in combination with the *ced-12* GAP allele lethality dropped significantly to 10% (Fig 7B). These results suggested the GAP activity of CED-12 acts on RHO-1, while the GEF function acts in parallel to RHO-1.

To test for interactions with CDC-42, we used mutations in the CDC-42 effector WSP-1. The null allele *wsp-1(gm324)* [48]showed 27% embryonic lethality due to arrests during ventral enclosure and elongation (23% and 4%, respectively) and combining *wsp-1* with *ced-12 (n3261-null)* increased lethality to 34% (n = 451). We could not combine the *wsp-1* mutation with a *ced-5* mutation since they are less than one map unit apart on the chromosome. The *ced-2(e1752)* allele combined with *wsp-1(gm324)* showed slightly elevated lethality compared to *wsp-1(gm324)*, 29%, that was not significant (n = 360). However, combining *wsp-1* with the *ced-12* GAP mutant reduced lethality significantly to 10% (n = 570) (Fig 7B). These results suggested the GAP activity of CED-12 acts on CDC-42, while the GEF function may act in parallel.

## Effects on levels of active Rac1/CED-10

Since a Rac1 biosensor does not exist for monitoring embryos, we instead measured effects on the Rac1 effector, GFP::WVE-1. Active Rac1/CED-10 is required to activate the WAVE

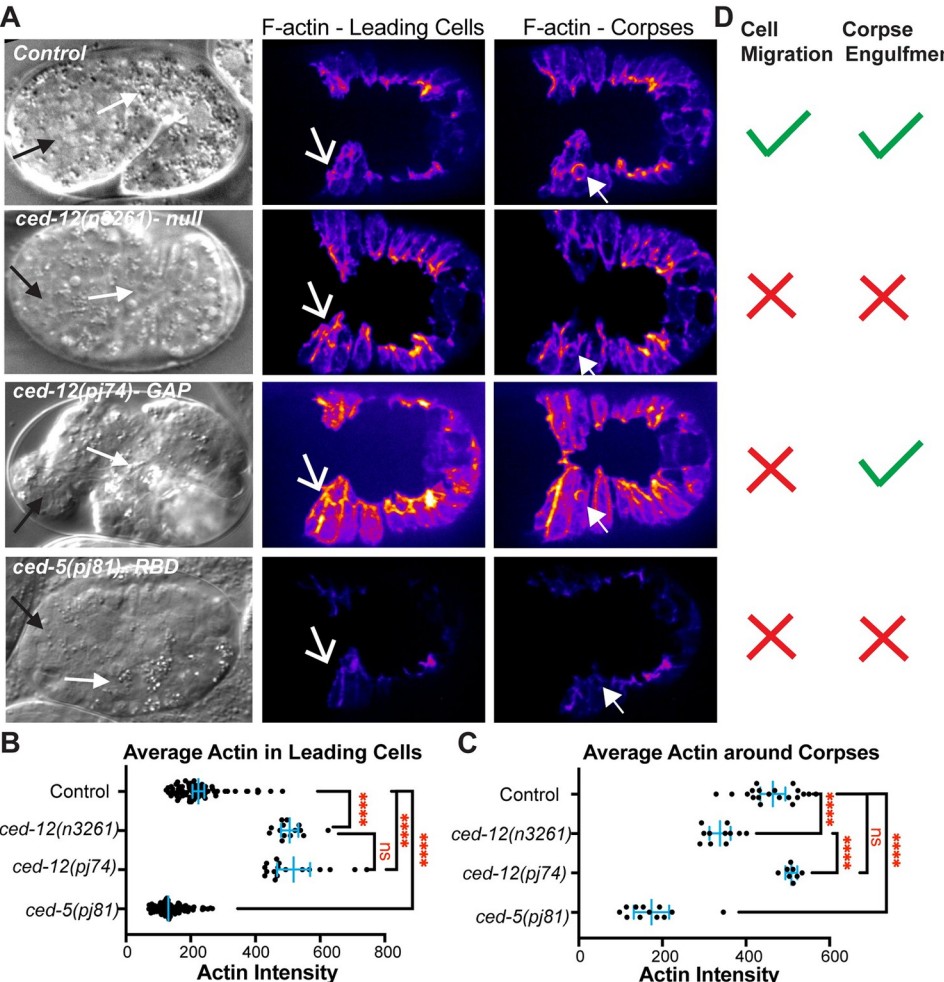

**Fig 6. CED-12 GAP mutant alters cell migration, not corpse engulfment, while CED-5 Rac binding domain mutant alters both.** (A) Embryos were imaged by DIC (left column) and using *lin-26p*::*LifeAct*::*mCherry* to visualize cell migration defects, corpse engulfment defects and F-actin levels. F-actin is shown during ventral enclosure. Left column: black arros point to anterior pharynx, white arrows point to anterior intestine, as in Fig 3. Center column: white open arrows point to migrating LCs. Right column: closed white arrows indicate C2 corpses in the epidermis. (B) Average intensity of F-actin in LCs during ventral enclosure was analyzed as in Fig 1B. (C) Average intensity of F-actin around C1, C2, and C3 corpses was measured as in Fig 4. (D) Summary of the phenotypes, regarding defects in epidermal cell migration, or in corpse engulfment, for each genotype. Asterisks denote statistical significance, **** = $p < 0.0001$, as determined by one-way Welch's ANOVA test.

complex [49]. If CED-5/CED-12 act as a GEF for CED-10/Rac1, loss of the GEF function should reduce levels of GFP::WVE-1. Interfering with Rac1 binding did not significantly change the levels. If CED-5/CED-12 act as a GAP for CED-10/Rac1, loss of the GAP function should increase levels of GFP::WVE-1. The levels did not increase and instead dropped in all tissues measured, including the epidermis (Fig 7C). This result suggested that the GEF function does not simply promote active Rac1, and suggested CED-12 is not a GAP for CED-10/Rac1.

## Effects on levels of active RHO-1/RhoA

Active RHO-1 in embryos can be measured with a biosensor that uses a region of ANI-1/anillin that binds to active RHO-1, *pie-1p*::*gfp*::*ani-1(AH+PH)* [50]. Loss of the GAP activity, using

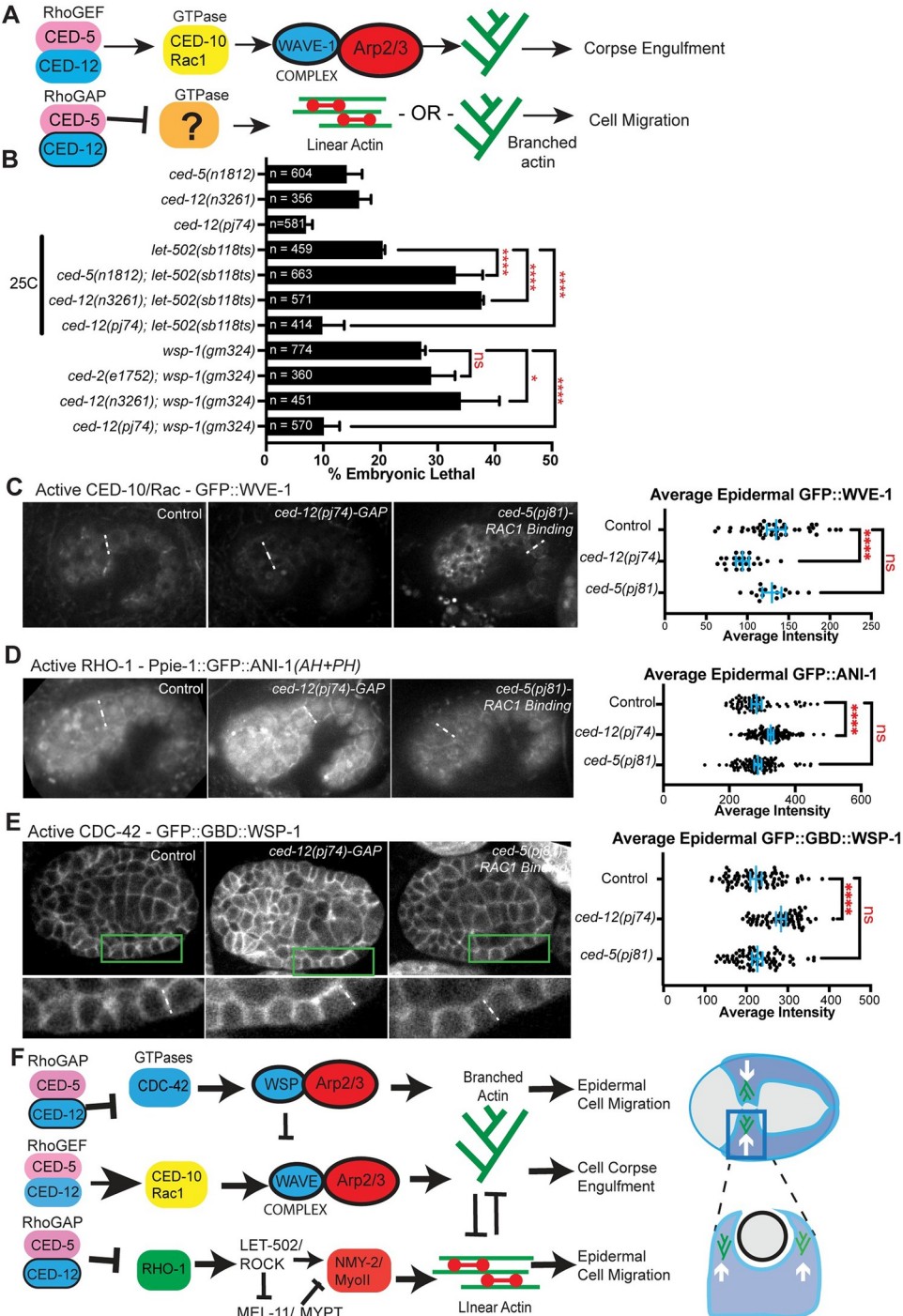

**Fig 7. The GAP function of CED-5/CED-12 may regulate RHO-1/RhoA and CDC-42.** (A) While the CED-5/CED-12 GEF function is known to active the GTPase CED-10/Rac1 during corpse engulfment, the GTPase target of the proposed GAP function of CED-5/CED-12, proposed to regulate cell migration, is not known. (B) Genetic analysis compared embryonic lethality of *ced-5*, *ced-2* and *ced-12* single mutants and double mutants with the RHO-1 pathway, using Rho Kinase mutant, *let-502(sb118ts)*, and with the CDC-42 pathway, using WASP mutant *wsp-1(gm324)*. *let-502* experiments were done at 25˚C since this is a temperature sensitive allele [47]. (C) Since a Rac1 biosensor is not available we used the Rac1 GTPase target, *gfp::wve-1* [22] to compare "Active Rac1" levels. The signal at epidermal seam cell boundaries (dotted line) was measured. Embryos are oriented anterior to the left and dorsal up, at the 1.5 fold stage. (D) A biosensor for active RHO-1, *Ppie-1::gfp::ani-1(AH+PH)*[50], was used to compare the effects of mutating the proposed GAP and Rac1-binding regions on CED-12 and CED-5. Levels of active RHO-1 at the epidermal seam

cell boundaries of 1.5 fold embryos (dotted line) were measured. Embryos are oriented as in C. (E) A biosensor for active CDC-42, *gfp*::*gbd*::*wsp-1* [51], imaged from a ventral view, anterior at left., allowed comparison of active CDC-42. Using the line scan function on ImageJ, average measurements were made at cell junctions in lateral epidermal cells, at approximately 320 minutes after first cleavage, 23˚C. Asterisks denote statistical significance, * = p<0.05, ** = p<0.01, **** = p<0.0001, as determined by one-way Welch's ANOVA test. (F) Model summarizing CED-5/CED-12's role as a GEF towards CED-10 during corpse and engulfment, and a GAP towards CDC-42 and RHO-1 during cell migration. The embryonic lethality and active GTPase levels contributed by mutations in the CED-5/CED-12 proposed GAP function (R357A) and Rac1-binding mutant (SR1541/1542AA) and the effects on epidermal and corpse F-actin (Figs 6 and 7) led us to propose this model: The GEF function, as expected, regulates CED-10/WAVE to support corpse engulfment. However, during epidermal cell migration, the role of CED-5/CED-12 is better explained as the result of the GAP function regulating branched and linear actin through the GTPases CDC-42 and RHO-1. Together these results illustrate how the bifunctional CED-5/CED-12 complex can support embryonic development through a bifunctional role on GTPase regulation.

*ced-12(pj74)* resulted in a significant increase in levels of active RHO-1, as seen by the increase signal at the junctions of the epidermal seam cells (Fig 7D). This result supported the genetic result that the GAP function regulates RHO-1 (Fig 7B). Loss of Rac1 binding using *ced-5(p81)* did not affect levels of active RHO-1. This result suggested that CED-5 is not a GEF for Rho.

*Effects on levels of active CDC-42*: Active CDC-42 in embryos can be measured with a biosensor that uses the GTPase binding domain of WSP-1, *cdc-42p*::*gfp*::*GBD*::*wsp-1* [51,52]. Loss of GAP activity using *ced-12(pj74)* resulted in a significant increase in levels of epidermal *cdc-42p*::*gfp*::*GBD*::*wsp-1*, whereas loss of Rac1 binding through *ced-5(pj81)* resulted in no significant change (Fig 7E). This result supported the genetic result that the GAP function regulates CDC-42.

Altogether these genetic and gene expression studies strongly suggest that the GAP activity of CED-5/CED-12 may regulate the GTPases RHO-1 and CDC-42.

## Discussion

### CED-5/CED-12 have previously undescribed role in embryonic morphogenesis

The discovery of CED-12 was impactful, since it identified a molecule, conserved across phyla, that affected cell migrations and cell engulfment of apoptotic corpses [13–16] (The cell migration defects referred to the larval migrations of the distal tip cell (DTC). These early papers hinted at a possible embryonic role [13]. However, the proposed embryonic role had not been further explored. Our findings here strongly support that the embryonic lethality of *ced-12*, also seen in partner protein *ced-5*, is due to failure to regulate F-actin during embryonic cell migrations, resulting in increased F-actin, altered dynamics and faster epidermal enclosure as the cells increase their migration rate.

The reason this embryonic role was mostly missed is that even with null alleles, the penetrance is relatively low, with only 9 to 16% of embryos dying (Table 1). This is in contrast to 100% embryonic lethality seen with complete loss of Rac1/CED-10, or loss of WAVE or Arp2/3 complex components [6,7]. Loss of *ced-5* or *ced-12* leads to partially penetrant lethality, and highly penetrant increase in F-actin in the migrating epidermis. These differences suggest two conclusions: (1) there is likely a second pathway, in addition to CED-5/CED-12, that regulates epidermal F-actin during this migration and (2) too much F-actin in the epidermal cells disrupts migrations. The *ced-5* and *ced-12* embryonic phenotype, with fully penetrant increased F-actin (100% of the embryos show elevated F-actin) and partially penetrant embryonic lethality (Figs 1 and 6 and Table 1) has been shown for other mutations in proteins that regulate actin. For example, loss of *hum-7/Myo9*, leads to increased F-actin in 100% of embryos, and 6% embryonic lethality [25,21].

Our results here showed additional roles for CED-5/CED-12 during embryonic development. Mutant *ced-5* and *ced-12* embryos were often shorter than normal, and reduced length correlated with reduced viability (Fig 3). Wild-type embryos are cuboidal, but once they enter the spermatheca and are fertilized, they are squeezed into an oval shape. The force for this may come from actomyosin in the spermatheca. Mutations in the *rho-1* pathway, like filamin, or the Rho GAP SPV-1 lead to embryos that are round [31,32], so too much or too little actomyosin contractility in the spermatheca can alter egg shape, including producing round embryos. Our results suggest that at these earliest stages of embryonic development, when oocytes are being fertilized, the branched actin regulators CED-5/CED-12, CED-10 and WAVE complex components, are required so that embryos are the correct length and shape, by promoting the correct levels of RHO-1 activity.

## CED-5/CED-12's proposed GAP function blocks excess F-actin to promote embryonic morphogenesis

Null mutations in CED-5 or CED-12 have consistently higher levels of F-actin in the migrating epidermal cells, and this correlated with faster migrations, and increased dynamics in the protrusions and retractions at the edge of leading cells (Figs 1 and 2). Surprisingly, the same epidermal cells that showed increased F-actin in the migrating cells, also show reduced F-actin around corpses (Figs 2 and 4). To untangle this puzzling dual nature of the CED-5/CED-12 GEF, we searched for new functions, and discovered them by comparing CED-12 to other ELMOs and ELMODs, and also to other GAP proteins. It has been shown that ELMODs mainly function as GAPs for Arf GTPases in processes such as protein traffic [36]. We identified a clear GAP domain within CED-12, that aligned to the GAP in ELMODs, and a related region in ELMOs, which do not contain the catalytic Arginine needed for GAP function (Figs 5A and S1A). Therefore, we tested if *C. elegans* uses CED-12 as a bifunctional ELMO/ELMOD.

We used the recently published structures of DOCK2/ELMO1/Rac1 and DOCK5/ELMO1/Rac1 to help us test the bifunctional hypothesis for CED-12 function. One idea that we considered, based on the structure, was that CED-12 may have two roles, depending on the open vs. closed conformation for the entire complex. For example, while the GAP residues point away from bound active Rac1, perhaps in the closed conformation CED-12 could help hydrolyze active Rac1 to its GDP state. Based on the published structures, this is not likely. The catalytic Arginine, R537, points away from the Rac binding domain in both open and closed conformations. This outward orientation of R537 actually increases the interactions that CED-12 can make, since the GAP motif is available in both open and closed conformations to interact with other proteins. This led us to test if CED-12 could couple the activation of one GTPase with the turn-over of a second GTPase.

## Evidence that two other GTPases are likely targets of the CED-12 GAP function

Genetic analysis and measurements of levels of biosensors for active GTPases suggested that CED-12 GAP function regulates RHO-1 and CDC-42 (Fig 7B–7E). Since the effect of mutating the CED-12 GAP function includes rescuing embryonic lethality (Fig 7B), this result suggests that excess levels of CDC-42 and RHO-1 are detrimental for embryonic epidermal morphogenesis. Our previous studies have identified GAPs that normally regulate RHO-1 and CDC-42, to prevent excess epidermal F-actin, and support embryonic morphogenesis and viability [25,21].

An alternative explanation for our results could be that the GEF activity of CED-5/CED-12 also activated the GTPases RHO-1 and CDC-42. We do not favor this interpretation. First,

DOCK GEFs have never been shown to activate RHO-1 [53–55]. Second, the structure for DOCK/ELMO/Rac1 argues against this. We threaded the structure of RHO1 and CDC-42 to replace Rac1 in the published structure (S1C Fig). As expected, this created structural interference with the key residue on CED-5/DOCK2 (R1542/M1529) that needs to interact with the ring residue of the GTPase (Rac1/CED-10 (W56), CDC-42(F56) and RHO-1(W58) (S1C Fig). While Rac1/CED-10 W56 can interact correctly with R1542, CDC-42 F56 and RHO-1/RhoA W58 are predicted to clash with M1529/R1542.

We tried to devise surgical changes that would only affect the GEF function, similar to how we removed the GAP function. This was not possible. Our first attempt to mutate the GEF function was to mutate the PXXPXXP domain at the C terminus of CED-12, proposed to support GEF function [13,56]. This mutation led to low level embryonic lethality (4%) and no change in epidermal F-actin levels (Table 1). We next tried to interfere with GEF function by mutating two residues that mediate binding to Rac1, based on the structures. This mutant, *ced-5(pj81)*, resembled null mutations in *ced-5* and *ced-12*, since it altered corpse engulfment and epidermal cell migration, with similar but increased levels of embryonic lethality. However, it had opposite effects on epidermal F-actin levels (lowering them) (Fig 6A and 6B). One explanation for why mutating these two Rac1 binding residues is worse than simply deleting *ced-5* is that this Rac1-binding mutation may trap active Rac1 molecules, acting as a Rac1 sink. We propose that other GEFs can activate Rac1, but in animals with the *ced-5(pj81)* mutations, other Rac1 GEFs may lose access to Rac1.

Loss of the CED-12 GAP function resulted in higher RHO-1 and CDC-42 biosensors, and lower GFP::WVE-1 used for a Rac Biosensor. With the caveat that we need a better Rac1 biosensor for *C. elegans*, this result may indicate cross talk between GTPases. Previous studies on GTPases show that cross talk between GTPases and their regulators make the interpretation of knock down studies complicated[57]. For example, the role of RhoA GAP CYK-4/MgcRac-GAP during cytokinesis is controversial. Some studies propose CYK-4 GAP supports cytokinesis by activating RhoA function, since CYK-4 binds to, localizes and activates the RhoA GEF ECT-2 at the equatorial plasma membrane [58,59]. Other studies propose CYK-4 Rho GAP supports cytokinesis by inactivating Rac1, since loss of Rac1 rescues loss of *cyk-4* [60]. However, Arp2/3, which does not localize to the contractile ring, nevertheless supports cytokinesis by blocking excess formin activity. Thus, the two populations of F-actin, linear (RhoA and formin-dependent) and branched (Rac1 and Arp2/3 dependent) compete for actin at the equatorial plasma membrane [61], which would require regulation of multiple GTPases.

To further investigate which GTPases are targets of the proposed GEF and GAP functions of CED-5/CED-12, it may be possible to test binding of the purified CED-5/CED-12 complex to GTP and GDP loaded versions of the three main GTPases, as we and others have done [46,25]. This will require working out the conditions for purifying the active complex. One possible result could be that purified Control CED-5/CED-12 complex binds to GTP loaded CDC-42 and RHO-1, but not CED-10-GTP, while purified CED-5/CED-12 complex with the GAP mutant shows loss of binding to CDC-42-GTP and RHO-1-GTP. Since the Rac1-binding mutant appears to be more than a simple loss of GEF activity, we cannot predict what it may show in such assays.

To incorporate all of these findings, we propose the following model for CED-5/CED-12 function (Fig 7E). To properly promote branched actin to engulf corpses, the GEF function of CED-5/CED-12 activates CED-10/Rac1, exactly as has previously been proposed. To regulate F-actin during embryonic epidermal cell migration, a different GEF must work to activate CED-10. There are 18 other GEFs found in *C. elegans*. Several have been shown to regulate embryonic development, mainly by regulating CDC-42 or RHO-1 [62, 51,63]. We are investigating GEFs that act on CED-10.

During epidermal cell migrations, CED-5/CED-12 have an important role limiting the activity of CDC-42, and this requires the proposed CED-12 GAP function. We previously published the surprising observation that in *wsp-1(gm324)* mutants, which are approximately 30% lethal, epidermal F-actin is too high, and the cells migrate too quickly [21]. This resembles what we show here for loss of the CED-12 GAP function. Perhaps in GAP mutants with excess CDC-42, too much WSP-1 is activated, which also promotes excessive branched F-actin. This suggests too much or too little WSP-1 has same phenotype: too much F-actin, and some dead embryos. Similar findings have been shown for regulators of the RHO-1 pathway [31,32].

During epidermal cell migrations, CED-5/CED-12 have an important role limiting the activity of RHO-1, and this requires the proposed CED-12 GAP function (Figs 6B, 6C, 6F and 7E). If RHO-1 is overly active, more actomyosin would form, driving more linear F-actin. It was possible that excess RHO-1 would increase membrane and cortical tension, thus inhibiting protrusions, as has been proposed [64]. However, increased epidermal F-actin correlated with increased leading edge membrane dynamics (Fig 2C), suggesting excess RHO-1 supported protrusions, perhaps through increased tension, as has been shown in some migrating cells (Reviewed in [65]). A second consequence of excess RHO-1, given that levels of actin can be limiting in some cell types [66], is reduced monomeric actin available to form branched actin. At this time, we do not have tools to specifically monitor branched actin vs. linear actin in these cells, so this prediction that excess RHO-1 leads to excess linear actin at the expense of branched actin, will have to await new tools.

How may one protein complex be regulated to both promote and inhibit F-actin? Our model predicts that different upstream regulators recruit CED-5/CED-12 for the two processes, epidermal cell migration vs. epidermal corpse engulfment. Attempts to tag CED-12 with GFP using CRISPR are underway. When we can image endogenous CED-12 in live animals, one prediction is that there will be different populations of CED-12, carrying out distinct functions, both promoting and inhibiting F-actin, in different parts of the epidermal cells.

## Materials and methods

### C. elegans Strains

All strains built for this study are listed in **Table 3**, and existing strains used are listed in **Table 4**, along with their relevant Reference. All strains were grown at 23°C unless otherwise stated.

### RNAi experiments

All RNAi bacterial strains used in this study were administered by the feeding protocol as in [22]. RNAi feeding experiments were done at 23°C unless otherwise mentioned. Worms were synchronized and transferred onto seeded plate containing RNAi-expressing bacteria. To monitor effectiveness of the RNAi we used two methods. We counted the percent dead embryos, which after three days is expected at >90% for *gex-3*. We also monitored post-embryonic silencing of a gfp-tagged strain in the intestine, such as *gfp*::*gex-3*. All RNAi treatments were done for three days.

### Live imaging

Imaging was done in a temperature-controlled room set to 23°C on a Laser Spinning Disk Confocal Microscope with a Yokogawa CSUX scan head, on a Zeiss AxioImager Z1 Microscope using the Plan-Apo 63X/1.4NA or Plan-Apo 40X/1.3NA oil lenses. Images were captured on a Hamamatsu CMOS Camera using MetaMorph software, and analyzed using

**Table 3. *C. elegans* strains built for this paper.**

| Strain | Genotype |
|---|---|
| OX764 | *ced-5(n1812); lin-26p::LifeAct::mCherry* |
| OX869 | *ced-12(n3261); lin-26p::LifeAct::mCherry* |
| OX924 | *ced-10(tm597)/nT1-gfp; lin-26p::LifeAct::mCherry* |
| OX799 | *ced-5(n1812); lin-26p::LifeAct::mCherry; dlg-1::gfp* |
| OX778 | *ced-12(k149); dlg-1::gfp* |
| OX775 | *ced-2(e1752); dlg-1::gfp* |
| OX293 | *ced-10(tm597)/nT1-gfp; dlg-1::gfp* |
| OX994 | *lin-26p::LifeAct::mCherry; smIs34[ced-1p::ced-1::gfp::rol-6(su1006)]* |
| OX1002 | *ced-5(n1812); lin-26p::LifeAct::mCherry; smIs34[ced-1p::ced-1::gfp::rol-6(su1006)]* |
| OX1003 | *ced-12(n3261); lin-26p::LifeAct::mCherry; smIs34[ced-1p::ced-1::gfp::rol-6(su1006)* |
| OX989 | *ced-12(pj74)* |
| OX1026 | *ced-5(pj81)* |
| OX991 | *ced-12(pj74); lin-26p::LifeAct::mCherry; dlg-1::gfp* |
| OX1046 | *ced-5(pj81); lin-26p::LifeAct::mCherry; smIs34[ced-1p::ced-1::gfp::rol-6(su1006)]* |
| OX1047 | *ced-5(n1812); let-502(sb118ts); OX1048 ced-12(n3261); let-502(sb118ts)* |
| OX1048 | *ced-12(n3261); let-502(sb118ts)* |
| OX1049 | *ced-12(pj74); let-502(sb118ts)* |
| OX1050 | *ced-2(e1752); wsp-1(gm324)* |
| OX1051 | *ced-12(n3261); wsp-1(gm324)* |
| OX1052 | *ced-12(pj74); wsp-1(gm324)* |
| OX1040 | *ced-5(pj81); gfp::wve-1* |
| OX1041 | *ced-12(pj74); gfp::wve-1* |
| OX1042 | *ced-5(pj81); cdc-42p::gfp::gbd::wsp-1* |
| OX1043 | *ced-12(pj74); cdc-42p::gfp::gbd::wsp-1* |
| OX1044 | *ced-5(pj81); pie-1p::gfp::ani-1(AH+PH)* |
| OX1045 | *ced-12(pj74); pie-1p::gfp::ani-1(AH+PH)* |
| OX776 | *ced-2(e1752); ced-12(k149)* |

**Table 4. Strains used in this paper.**

| Strain | Genotype | Reference |
|---|---|---|
| JUP38 | *lin-26p::LifeAct::mCherry* | [20] |
| FT48 | *xnIs16[dlg-1::gfp]* | [23] |
| FT1459 | *xnIs506 [cdc-42p::gbd-wsp-1::gfp]* | [51] [52] |
| MG617 | *xsSi5[pie-1p::gfp::ani-1(AH+PH)::pie-1 3'UTR + Cbr-unc-119(+)]* | [50] |
| HR1157 | *let-502(sb118ts)* | [47] |
| NG324 | *wsp-1(gm324)* | [48] |
| MT4434 | *ced-5(n1812)* | [19] |
| MT11068 | *ced-12(n3261)* | [16] |
| NF87 | *ced-12(k149)* | [13] |
| CB3257 | *ced-2(e1752)* | [5] |
| MT5013 | *ced-10(n1993)* | [5] |
| LE198 | *ced-10(tm597)/nT1-gfp* | [6] |
| OX669 | *pj64 [gfp::3xflag::wve-1]* | [22] |
| OX466 | *pjIs4[ced-10p::ced-10::gfp]* | [3] |

ImageJ. Controls and mutants were imaged within 3 days of each other with the same imaging conditions. All measurements were performed on raw data using ImageJ. For fluorescent measurements, background intensity was subtracted by using a box or line of the same size and measuring average intensity in the same focal plane, near the animal.

## Quantitation of immunofluorescence

Quantitation of live fluorescence was performed using the line selection and the dynamic profile function of ImageJ to measure fluorescence along lines of equal lengths. For all experiments shown, the images were captured at the same exposure settings for wild type and mutants. All quantitation was done on the raw images. The figure legends indicate when images were enhanced for contrast, and the same enhancement was applied to a mosaic of the related images for that experiment. Each measurement was taken following the subtraction of background fluorescence.

## Statistical analysis

For grouped data, statistical significance was established by performing a one-way Analysis of Variance (ANOVA), the Brown-Forysythe and Welch ANOVA, followed by a Dunnett's multiple comparisons T3 post-test. For ungrouped data, an unpaired t-test, the unequal variance (Welch) t test, was used. Error bars show 95% confidence intervals. Asterisks (*) denote p values * = p < .05, ** = p<0.001, *** = p<0.0001, **** = p<0.00001. All statistical analysis was performed using GraphPad Prism 8.

*UCSF*: https://www.rbvi.ucsf.edu/chimerax

Modeling CED-5/CED-12 with and without CED-10: CED-5/CED-12 were modeled bound to CED-10 and in its apo form using DOCK2/ELMO1/Rac1 and DOCK2/ELMO1 coordinates (PDB IDs: 6TGC and 6TGB). The models were generated using SWISS-MODEL, and visualized in UCSF Chimera.

## Supporting information

**S1 Fig.** (A) Aligning CED-12 to the *C. elegans* GAPs CYK-4, SYD-1, RGA-1 and RGA-2 (top) identified a GAP region in CED-12. The catalytic arginine of the GAPs and CED-12 is shown in boldface and highlighted. Aligning CED-12 with human ELMOs shows a similar GAP-like region exists in ELMOs. Aligning CED-12 with ELMODs from Hs (human), Dm (Drosophila), Aq (Amphimedon queenslandica) or Bd (Batrachochytrium dendrobatidis) (bottom) illustrated they have a conserved region, that includes the arginine residue R537, boxed in red in all alignments. (B). CED-5 aligned to DOCK2 identified a conserved Rac1-binding domain. The SR mutated to AA in *ced-5(pj81)* is indicated. (C). DOCKs are thought to bind to Rac1 or CDC-42, but not RHO-1/RhoA. Models compare how Rac1/CED-10 W56 vs. CDC-42 F56 vs. RHO-1 W58 fit into the pocket of CED-5 S1541/R1542. See also Fig 5.
(TIF)

**S1 Movie. *ced-12* mutants have elevated epidermal F-actin and enclose faster.** Images of embryos undergoing ventral enclosure, made every minute for 15 minutes. Epidermal F-actin in live embryos was imaged using the *lin-26p::LifeAct::mCherry* strain [20], pseudo-colored with ImageJ Fire. Embryos are ventral up and anterior to the left. Left: Control embryo. Center: *ced-12(n3161)* null embryo. Right: *ced-12(pj74)* GAP mutant. Exposures are set equally, and embryos are timed equally. Note that protrusions and retractions occur along the length of the migrating epidermal cells. See also Figs **1B, 1C** and **2C**.
(AVI)

**S2 Movie. Comparison of *ced-12(n3261)* embryos that live and that die.** All *ced-12(n3261)* embryos express high levels of F-actin, monitored with *lin-26p*::*LifeAct*::*mCherry* (see also **Fig 1B**). However, 84% of these embryos enclose and live, while 16% arrest and die. Embryos that arrest tend to be shorter. See also **Fig 3**. Note that protrusions and retractions occur along the length of the migrating epidermal cells in embryos that live and in embryos that die. For close ups of Protrusions and Retractions, see also **Fig 2C**.
(AVI)

**S1 Raw Data. Supplemental Raw Data for all Figures.** Excel file with pages to match each Figure and Table are included in one file. Individual Excel pages correspond to each Table and graph included in the text and in the main Figures.
(ZIP)

## Acknowledgments

We thank the NCRR-funded *Caenorhabditis* Genetics center (CGC), funded by NIH Office of Research Infrastructure Programs (P40 OD010440), for strains. Molecular graphics and analyses performed with UCSF Chimera, developed by the Resource for Biocomputing, Visualization, and Informatics at the University of California, San Francisco, with support from NIH P41-GM103311. We thank members of the Soto Lab and William Wadsworth for comments.

## Author Contributions

**Conceptualization:** Martha C. Soto.

**Data curation:** Thejasvi Venkatachalam, Martha C. Soto.

**Formal analysis:** Thejasvi Venkatachalam, Sushma Mannimala, Yeshaswi Pulijala, Martha C. Soto.

**Funding acquisition:** Martha C. Soto.

**Investigation:** Sushma Mannimala, Yeshaswi Pulijala.

**Resources:** Martha C. Soto.

**Supervision:** Martha C. Soto.

**Validation:** Martha C. Soto.

**Visualization:** Thejasvi Venkatachalam, Sushma Mannimala, Yeshaswi Pulijala.

**Writing – original draft:** Martha C. Soto.

**Writing – review & editing:** Thejasvi Venkatachalam, Martha C. Soto.

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
