## [Decision Letter · Decision Letter 0]

4 Dec 2023

Dear Dr Soto,

Thank you very much for submitting your Research Article entitled 'CED-5/CED-12 (DOCK/ELMO) can promote and inhibit F-actin formation via distinct motifs that target different GTPases' to PLOS Genetics.

The manuscript was fully evaluated at the editorial level and by independent peer reviewers. The reviewers appreciated the attention to an important problem, but raised some substantial concerns about the current manuscript. Based on the reviews, we will not be able to accept this version of the manuscript, but we would be willing to review a much-revised version. We cannot, of course, promise publication at that time.

Should you decide to revise the manuscript for further consideration here, your revisions should address the specific points made by each reviewer. We will also require a detailed list of your responses to the review comments and a description of the changes you have made in the manuscript.  The reviewers have various suggestions for experiments to solidify the conclusions, and we would expect that at least some of these suggestions be taken up. Other concerns may be addressed by toning down the strength of the claims in the paper.

If you decide to revise the manuscript for further consideration at PLOS Genetics, please aim to resubmit within the next 60 days, unless it will take extra time to address the concerns of the reviewers, in which case we would appreciate an expected resubmission date by email to plosgenetics@plos.org.

We are sorry that we cannot be more positive about your manuscript at this stage. Please do not hesitate to contact us if you have any concerns or questions.

Yours sincerely,

Andrew D. Chisholm

Academic Editor

PLOS Genetics

Gregory P. Copenhaver

Editor-in-Chief

PLOS Genetics

Reviewer's Responses to Questions

**Comments to the Authors:**

Reviewer #1: The study from the Soto lab reported distinct roles of CED-5/CED-12 (DOCK/ELMO) in actin assembly during apoptotic cell clearance and embryonic cell migration. It has been known that two component GEF, CED-5/CED-12 functions as the GEF of CED-10/Rac1 during apoptotic cell engulfment; however, their potential roles in cell migration remains unclear. This study reported that CED-5/CED-12 (DOCK/ELMO) plays roles in embryonic cell migration, which is not a surprising results. However, they found that the loss of CED-5/CED-12 led to an increased level of F-actin in the leading edge and the membrane becomes more dynamic (data quality comments below). The author suggests that CED-5/CED-12 may inhibit actin assembly in the migrating epidermis. Further, they did some residue replacement experiments to suggest that CED-12 may act as GAP. Their data further suggests that other small GTPase such as RHO-1 and CDC-42 might be involved. While it is possible that CED-5/CED-12 may indeed play distinct roles of different events, they need obtain more solid evidence to support their major conclusion. Alternatively, they can tone down their conclusions.

First of all, visualizing F-act amount is not very trivial. Although different markers have been established, the field also pointed out various caveats using these marker. Usually, if the established marker reveals a finding that is consistent with the previous reports, it can be fine. Otherwise, a second marker should be used to backup their conclusion. In this scenario, the author can use the fluorescence phallodin to re-examine the upregulation of F-actin amount. This is a crucial experiment and should be easily performed in C. elegans embryos.

Following the above point, the Figure 1E is not very supportive. How did the author measure the leading cell dynamics? Is the leading cell or the leading edge of a migrating cell? It seems that there is a small increase; however, it is indeed quite subtle. Can the author measure how much increase? How does this level increase correlate with increased F-actin amount? The author should present image frames from time-lapse movies to show the dynamics.

Considering that fluorescence Arp-2/3 complex, WSP-1 and WAVE have already been deposited in the CGC, the author should examine if these actin nucleation factor also upregulated in the leading edge.

While not very necessary but definitely helpful, the biochemical analysis of the GEF or GAP activity should be presented or discussed.

The title is quite confusion: can the author specify when promote and when inhibit? If promotion is known, the author can emphasize the novel findings.

The writing is too lengthy. Many descriptions in the introduction have already been in the textbook.

The author should follow C. elegans gene/ protein nomenclature: it is CED-5.

Reviewer #2: The small GTPase Rac/CED-10 is known to be required for ventral epidermal cell migration during embryonic morphogenesis, but the Guanine Nucleotide Exchange Factor (GEF) activating it has not been characterized so far. The authors ask whether the bipartite Rac-GEF CED-5/CED-12 could fulfil this role. They find that strong mutations in those proteins affect the speed of ventral enclosure and lead to an increase in actin dynamics at the leading edge, arguing that they do present the hallmarks of proteins being required for ventral enclosure. However, those ced-5 or ced-12 mutations do not enhance the lethality of a partial ced-10 mutation. In search for an explanation for this conundrum, they remark that CED-12 presents some discreet homology to the active site of Rho GTPase Activating Proteins (RhoGAP). They find that mutating a critical Arg residue in this putative CED-12 GAP motif behaves as a partial suppressor of the RhoA effector Rho-kinase/LET-502, and that null mutations in ced-5 or ced-12 enhance a let-502 hypomorph. Likewise, the CED-12 GAP mutation suppresses a mutation in the Cdc42 effector WSP-1, suggesting that CED-5/CED-12 might act through Rho or Cdc42. Finally, they find that the mutation in the putative CED-12 GAP domain enhances actin level at the leading edge of epidermal cells during ventral enclosure, whereas a mutation in CED-5 within a domain with partial homology to the Rac-binding domain of the CED-5 vertebrate homolog DOCK2 removes actin from the leading edge.

They propose that CED-5/CED-12 would act as a bona fide RACGEF in apoptosis, but not during ventral enclosure. Instead, they argue that it acts as a GAP for Cdc42 and Rho to limit the amount of linear actin.

The genetic results are in general compelling, although the differences are sometimes limited such as the enhancement of wsp-1(gm324) lethality by ced-5/12, or only double lethality such as the effect on let-502(sb118), both of which are slightly indirect and the change in lethality might be significant only at another stage than ventral enclosure; the interactions with Rho-1(RNAi) or Cdc42(RNAi) are also contradictory as to the role of CED-5 vs CED-12 on RhoA. So, I take positively their genetic arguments, but I am not yet convinced by the broader conclusions. The authors argue that the putative CED-12 GAP domain is unlikely to come in contact with Rac/CED-10 owing to its position in the structure. Given the homology between Rac, Cdc42 and RhoA, how would CED-5/CED-12 be more prone to regulated Cdc42 and RhoA? Additionally, isn’t it surprising if the putative CED-5/12 GAP domain regulates filopodia inducing GTPases that there aren’t filopodia visible during ventral enclosure?

Major points before the work can be accepted:

1. The authors should model Cdc42 and RhoA with CED-5/12 as they did in Fig. 4 with Rac.

2. They should make sure that the enhancing or suppression lethality effects described in Fig. 5 are due to ventral enclosure defects, rather than any other embryonic defect involving the two players being tested.

3. They should explain why there is no apparent filopodia visible in Fig. 6A. If CED-5/CED-12 acts in pocket cells in the second step of ventral enclosure, which involves non-muscle myosin II and presumably RhoA/Rhokinase, then they should demonstrate changes in the level of actomyosin in those cells.

4. It was not entirely clear to me what is the explanation for the fact that both ced-12(pj74)-GAP and ced-5(pj76)-GEF both increase RhoA in the pharynx as evidenced from the ANI-1(AH-PH) marker.

Minor:

1. Figure 2B-D: the median value in grey is difficult to spot; please consider another colour.

2. Figure 3A-B: my eye looking at the ced-5 and gex-3 RNAi images of actin is disagreeing with the quantification displayed in panel B, inasmuch I see lower actin signal for ced-5(n1812) compared to gex-3. My eye may be fooled, but perhaps the example for ced-5 or gex-3 does not reflect the average. Please revisit.

3. Page 7, line 19: syntax problem for “and did loss” which is lacking a subject or some other word.

4. Page 9, line 4/5: it is let-502(sb118), not sb1008.

5. Please display the strain list in a separate Table, rather than a text. It will make it more readable.

Reviewer #3: Knowing that CED-12 and its partner CED-5 function as a GEF for CED-10/Rac1 to promote corpse engulfment of dying cells during embryogenesis and distal tip cell migration in C. elegans, Venkatachalam et al. characterize the role of CED-5/CED-12 during embryonic epidermal cell migration. The authors found that inhibition of CED-5/CED-12 led to some embryonic lethality but, counterintuitively, resulted in increased F-actin levels and cell migration velocity. To address how CED-

12/CED-5 could inhibit F-actin in this cellular context, they investigated the role of a GAP domain that has not been described previously. Mutating

a candidate catalytic Arginine in the CED-12 GAP region (conserved and usually mutated to inhibit other GAPs) resulted in decreased embryonic lethality when compared to the null cdc12 mutant, and specifically increased F-actin levels in epidermal migrating cells but not in engulfing corpses. Mutating a candidate GEF region on CED-5 predicted to bind

and stabilize Rac1 for GTP transfer, resulted in apparent failure in both corpse engulfment and epidermal cell migration. By doing genetic experiments and assessing levels of reporters for 3 GTPases, the authors propose a model where CED-5/CED-12 function both as a GEF and a GAP, supporting the cycling of multiple GTPases that may vary depending on the cellular context.

Although the main findings are interesting and novel, the manuscript and experimental work therein could be more rigorous, more thorough and better explained. It is my opinion that the manuscript should be significantly improved before considering it for publication in PLoS Genetics.

Below follows a list of comments/suggestions for improvement.

Major comments:

The mutants that were generated were based on predictions. What validates that they really correspond to a CED-12 GAP mutant and a CED5 GEF mutant?

An immunoblot to show protein levels in the several mutants could be shown. If the mutated protein is not expressed at wild-type levels, care has to be taken when taking conclusions.

Figure 5B: Explain to the reader why would the results obtained with a GAP null be different from a GAP point mutant.

Figure 5B: Should add the GEF point mutant (ced5(pj76)) in addition to the ced5 null.

Figure 5C: Effect of AHPH Rho biosensor should be assessed during epidermal cell migration - why looking at pharynx and buccal activity?

Effect of cdc42 biosensor should be assessed during epidermal cell migration

Measurements with GFP::WVE-1 - increase N and assess during epidermal cell migration

Opposite messages: In the text it says that “…while depleting rho-1 in ced-12(pj74) GAP mutant reduced lethality to

8%, and this drop was significant (Table 3).” In table 3: ced-12(pj74); rho-1 RNAi 0.76 lethality”

“cross the candidate GAP to hypomorphic alleles, or partial loss of function, of the target

GTPases. The prediction is that loss of a GAP in combination with a hypomorphic allele (ADD of the small GTPAse) will

rescue the loss of function phenotype… loss of a GEF in combination with a hypomorphic allele (ADD of the small GTPAse) will synergistically enhance the loss

of function phenotype.” - However, loss of let502 or wsp-1 brought embryonic lethality back to the mutants alone and did not improve it further. Explain.

“Neither loss of the GAP function, nor the GEF function increased the levels of GFP::WVE-1. Instead, loss of the GAP function using ced-12(pj74) resulted in significantly reduced gfp::wve-1 levels in all tissues measured, including in the nerve ring (Fig. 5E). ced-5(p76) resulted in no significant

change. This result further supported that the GEF function does not simply promote active Rac, and suggested CED-12 is not a GAP for CED-10/Rac1” -> if ced5 is a GEF for Rac1, shouldn’t WAVE signal decrease in the cdc5 GEF mutant? If CED-12 is not a GAP for Rac, then levels of the Rac1 reporter should not be affected; why are they decreased?

Add p values to all tables

Should quantify defects in cell migration velocity for the GAP mutant

ced5 GEF mutant - quantification of corpse engulfment o

---

## [Decision Letter · Decision Letter 1]

3 Jun 2024

Dear Dr Soto,

We are pleased to inform you that your manuscript entitled "CED-5/CED-12 (DOCK/ELMO) can promote and inhibit F-actin formation via distinct motifs that may target different GTPases" has been editorially accepted for publication in PLOS Genetics. Congratulations!

Please note that the reviewer has made recommendations for minor changes that we believe are reasonable. You should be able to attend to them as you prepare the final draft for the production team (the editorial team will not need to re-evaluate).

Yours sincerely,

Andrew D. Chisholm

Academic Editor

PLOS Genetics

Gregory P. Copenhaver

Section Editor

PLOS Genetics

Comments from the reviewers (if applicable):

Reviewer's Responses to Questions

**Comments to the Authors:**

Reviewer #2: I am satisfied with the authors’ responses to my comments, and I think their responses to the other reviewers’ concerns are satisfactory as well. So I support publication of this work to PLOS Genetics as it reports novel finding, and that the complexity of the phenotypes have been well studied. But first, please attend to the minor comments below (pages refer to the ms itself, not to the PDF including the rebuttal letter).

1. P2, line 25: the wording “cloned mutations in the Arp2/3 regulators, gex-2 and gex-3,” is incorrect. I guess the authors mean they identified the molecular nature of the genes mutated by mutations gex-2() and gex-3 - cloned remains a bit lab jargon.

2. Fig. 2A: I’m not convinced that the magnified pictures of CED-10::GFP (BTW, wouldn’t capital letters be preferable on the picture for proteins) are showing epidermal cells, as stated in the legend, since the cells look quite small and one can guess the pharynx outlines.

3. P5, sentence “Comparing wild type with ced-5 (n1812), ced-12(n3261) and gex-3(RNAi) showed that lengths were significantly shorter than wild type in the majority of mutant embryos, with the exception of ced-12(n3261) viable embryos (Fig. 3A,B).”, the wording could be streamlined to avoid repeating “wild-type”.

4. P7, line 30: the use of the expression “ventral engulfment” when refering to ventral migration or ventral enclosure is confusing, and should be avoided.

5. Fig. S1: label CED-12 with capital letters at the bottom of the alignment with ELMODs.

6. P9, line 34: when refering to “the three GTPases”, add of the Rho/Rac/Cdc42 family (there are many more GTPases in a cell).

7. P10, line 2: as indicated in the original review, it is let-502(sb118ts), not sb1118 nor sb1008.

8. P13, lines 12-13: when citing the effect of CYK-4 on Rho, be careful that one paper has suggested that CYK-4 works through Rac during cytokinesis (see PMID: 28298491).

9. Movie S1 and S2: Could you move the genotypes slightly down as moving the mouse over the window partially hides them

**Have all data underlying the figures and results presented in the manuscript been provided?**

Reviewer #2: Yes

PLOS authors have the option to publish the peer review history of their article (what does this mean?). If published, this will include your full peer review and any attached files.

Reviewer #2: **Yes: **Michel Labouesse

**Data Deposition**

http://datadryad.org/submit?journalID=pgenetics&manu=PGENETICS-D-23-01178R1

**Press Queries**

---

## [Editor Report · Acceptance letter]

16 Jul 2024

PGENETICS-D-23-01178R1 

CED-5/CED-12 (DOCK/ELMO) can promote and inhibit F-actin formation via distinct motifs that may target different GTPases 

Dear Dr Soto, 

We are pleased to inform you that your manuscript entitled "CED-5/CED-12 (DOCK/ELMO) can promote and inhibit F-actin formation via distinct motifs that may target different GTPases" has been formally accepted for publication in PLOS Genetics! Your manuscript is now with our production department and you will be notified of the publication date in due course.

With kind regards,

Livia Horvath

PLOS Genetics

On behalf of:
